# Mild myelin disruption elicits early alteration in behavior and proliferation in the subventricular zone

Elizabeth A Gould[1,2,3], Nicolas Busquet[4], Douglas Shepherd[5,6], Robert M Dietz[7], Paco S Herson[7], Fabio M Simoes de Souza[8], Anan Li[9], Nicholas M George[1,2,3], Diego Restrepo[1,2,3]*, Wendy B Macklin[1,2,3]*

[1]Department of Cell and Developmental Biology, University of Colorado Anschutz Medical Campus, Aurora, United States; [2]Rocky Mountain Taste and Smell Center, University of Colorado Anschutz Medical Campus, Aurora, United States; [3]Neuroscience Program, University of Colorado Anschutz Medical Campus, Aurora, United States; [4]Department of Neurology, University of Colorado Anschutz Medical Campus, Aurora, United States; [5]Department of Pharmacology, University of Colorado Anschutz Medical Campus, Aurora, United States; [6]Pediatric Heart Lung Center, University of Colorado Anschutz Medical Campus, Aurora, United States; [7]Department of Anesthesiology, University of Colorado School of Medicine, Aurora, United States; [8]Center of Mathematics, Computation and Cognition, Federal University of ABC, Brazil; [9]Jiangsu Key Laboratory of Brain Disease and Bioinformation, Research Center for Biochemistry and Molecular Biology, Xuzhou Medical University, Xuzhou, China

*For correspondence:
Diego.Restrepo@ucdenver.edu (DR);
wendy.macklin@ucdenver.edu (WBM)

Competing interests: The authors declare that no competing interests exist.

**Abstract** Myelin, the insulating sheath around axons, supports axon function. An important question is the impact of mild myelin disruption. In the absence of the myelin protein proteolipid protein (PLP1), myelin is generated but with age, axonal function/maintenance is disrupted. Axon disruption occurs in *Plp1*-null mice as early as 2 months in cortical projection neurons. High-volume cellular quantification techniques revealed a region-specific increase in oligodendrocyte density in the olfactory bulb and rostral corpus callosum that increased during adulthood. A distinct proliferative response of progenitor cells was observed in the subventricular zone (SVZ), while the number and proliferation of parenchymal oligodendrocyte progenitor cells was unchanged. This SVZ proliferative response occurred prior to evidence of axonal disruption. Thus, a novel SVZ response contributes to the region-specific increase in oligodendrocytes in *Plp1*-null mice. Young adult *Plp1*-null mice exhibited subtle but substantial behavioral alterations, indicative of an early impact of mild myelin disruption.
DOI: https://doi.org/10.7554/eLife.34783.001

## Introduction

Myelin-producing cells of the central nervous system, oligodendrocytes, facilitate normal neuronal function. While myelin acts as an insulator, it has additional roles crucial for axonal activity. Myelin disruption contributes to a growing list of neurological diseases, including the most common neurological disease in young adults, multiple sclerosis (MS)(*Browne et al., 2014*). Patients with extensive demyelination, such as in MS, exhibit severe motor deficits, as well as psychiatric symptoms, including a higher incidence of cognitive dysfunction (*Feinstein et al., 2013*). While psychiatric diseases are not typically associated with overt demyelination, subtle white matter abnormalities have been

observed (*Mighdoll et al., 2015*), and it is proposed that mild myelin alterations reduce conduction velocity, which alters timing within neural circuits, including oscillations (*Filley and Fields, 2016*; *Pajevic et al., 2014*; *Almeida and Lyons, 2017*).

Proteolipid protein (PLP1) is a major structural component of myelin (*Griffiths et al., 1998*) and mutations in the *Plp1* gene are deleterious in humans. *Plp1* point mutations or gene duplications lead to the dysmyelinating diseases, Pelizaeus-Merzbacher disease (PMD) and spastic paraplegia type 2 (*Torii et al., 2014*). Patients with mutations resulting in complete deletion of the PLP1 protein exhibit mild early impairments that progress to spastic quadriplegia, ataxia, and cognitive impairments in adolescence. Genetic deletion of PLP1 in mice does not impair myelination and induces subtle alterations in myelin structure that correlate with a slight reduction in spinal cord conduction velocity (*Klugmann et al., 1997*; *Petit et al., 2014*). However, by 16 months (16M) *Plp1*-null mice exhibit severe motor deficits, myelin loss and progressive axonal degeneration (*Griffiths et al., 1998*), indicating that PLP1 plays a role in myelin support of axonal integrity.

This study evaluates early cellular alterations contributing to progressive pathology in *Plp1*-null mice. To observe region-specific cellular alterations, we used high throughput methods to generate unbiased regional analysis. These analyses revealed previously unidentified cellular responses to mild myelin disruption. We provide evidence that enhanced proliferation in the subventricular zone (SVZ) leads to a progressive accumulation of new oligodendrocytes in the corpus callosum (CC) and olfactory bulb (OB). Recent studies of different mouse models suggest that mild myelin deficits eventually lead to cognitive deficits (*Poggi et al., 2016*), but our studies are the first to demonstrate mild myelin disruption resulting in specific behavioral alterations in young adulthood.

## Results

### Increased oligodendrocyte density in *Plp1*-null mice

To quantify oligodendrocytes, we studied *Plp1*-eGFP mice that express eGFP under the PLP1 promoter (*Mallon et al., 2002*). Oligodendrocyte distribution appeared grossly normal in *Plp1*-eGFP + *Plp1* null mice (*Figure 1A,B*). To provide high-volume analysis of oligodendrocytes, we optically cleared *Plp1*-eGFP tissue (*Mallon et al., 2002*), with a modified process by which eGFP was maintained in the cytoplasm of *Plp1*-eGFP+ cells (see Materials and methods) (*Yang et al., 2014*). *Plp1*-eGFP+ cell bodies and processes were visible in the cleared tissue (*Figure 1C,D*), and a comprehensive survey of changes in *Plp1*-eGFP+ cell number by cleared tissue digital scanned light-sheet microscopy (C-DSLM)(*Ryan et al., 2017*) suggested an increase in *Plp1*-eGFP+ oligodendroocytes in 6M *Plp1*-null OB and a significant increase in oligodendrocytes in the forebrain (*Figure 1E*, *Table 1*).

To validate the increased oligodendrocyte density in the *Plp1*-null mouse, we developed a semi-automatic quantification method to compare oligodendrocyte density in specific brain areas (*Figure 1—figure supplement 1*). This analysis was performed on tiled confocal images of 30 µm sagittal tissue sections. At 6M, *Plp1*-eGFP+ oligodendrocyte density was increased in the rostral CC/ genu and the OB of the *Plp1*-null brain, but not other regions (*Figure 1E–H*, *Figure 1—figure supplement 2*). The oligodendrocyte cell densities determined by C-DSLM vs confocal quantification were remarkably consistent (*Figure 1E,F*). Using the same quantification method, the oligodendrocyte density increase was confirmed with the pan oligodendrocyte marker, Olig2, and an increase in Olig2+cells was observed in the rostral CC prior to increased *Plp1*-eGFP+ cells (*Figure 1I,J*). A low-grade inflammatory response in the CC was demonstrated by increased GFAP immunoreactivity in 2M *Plp1*-null mice (*Figure 1—figure supplement 3*) (*Petit et al., 2014*).

### Accumulation of new oligodendrocytes in *Plp1*-null mice

We tested the prediction that progenitor proliferation resulted in the increase in oligodendrocytes (*Figure 2*). The thymidine analog EDU was injected and 3 weeks later, the density of EDU + SOX10+ (*Figure 2*, *Table 2*) and EDU + *Plp1*-eGFP+ oligodendrocytes in the CC and OB (*Figure 2D,E*, *Table 2*) was increased in 4M *Plp1*-null mice. TUNEL staining of the tissue was unchanged, suggesting normal cell death (*Figure 2—figure supplement 1*). Thus, the increase in oligodendrocytes in *Plp1*-null mice resulted from increased proliferation. The newly produced oligodendrocytes differentiated into Gst-pi+ mature oligodendrocytes (*Figure 2—figure supplement 2*).

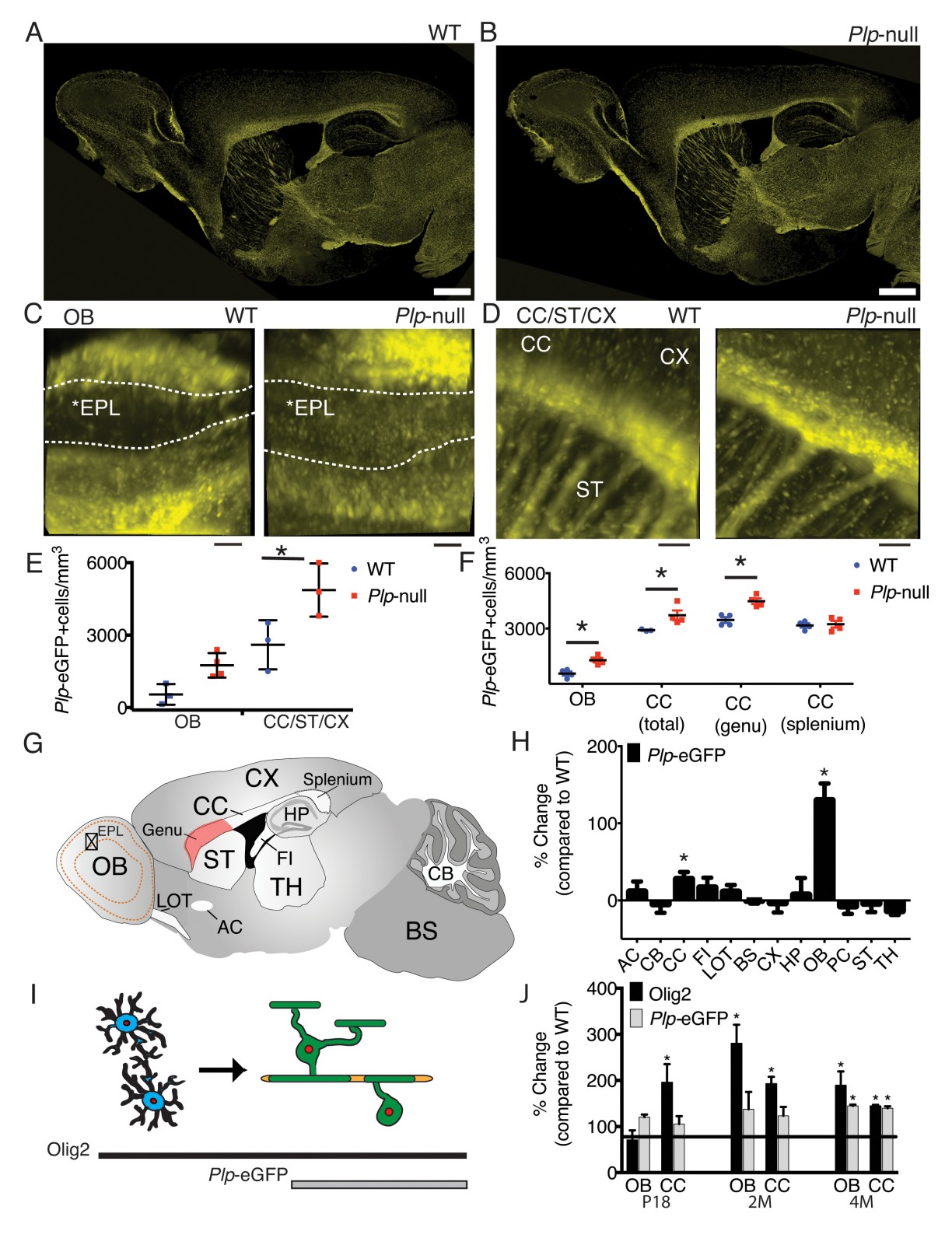

**Figure 1.** Region-specific increase in oligodendrocyte density in *Plp1*-null mice. (**A-B**) Confocal image of *Plp1*-eGFP expression in 6M WT and *Plp1*-null. Bar = 1 mm. C and D. *Plp1*-eGFP imaged with cleared tissue digital scanned light-sheet microscopy (C-DSLM) in cleared brain tissue. (**C**) *Plp1*-eGFP in cleared WT and *Plp1*-null OB. EPL is outlined. Bar = 50 μm. (**D**) *Plp1*-eGFP in cleared cortex. Bar = 100 μm. Image plane extracted from 1.3 × 1.3 × 6 mm axial image stack. Acquired using 10 μm exciting light-sheet, 10x/0.28 detection objective with 0.65 μm in-plane and 1 μm axial spacing. (**E**) Cell

*Figure 1 continued on next page*

*Figure 1 continued*

density in cleared tissue imaged with C-DSLM was increased in the CC of the 6M *Plp1*-null (F(1,9) = 15, 2-way ANOVA; p<0.01). (F) MATLAB quantification of confocal images reveals an increase in *Plp1*-eGFP cells in the OB (1283 ± 122 vs 558 ± 112 cells/mm$^3$; p<0.01) and CC (genu: 4483 ± 155 vs 3462 ± 152 cells/mm$^3$, p<0.01) of *Plp1*-null mice n = 4/genotype. (G) Brain regions (*Table 1*). (H) Percent change in cell number in 6M *Plp1*-null vs. WT. (I) Olig2 *Plp1*-eGFP in the oligodendrocyte lineage. (J) Olig2+ cells were increased in P18 CC (195 ± 39% of WT), 2M OB (280 ± 40%) and CC (192 ± 15%), and 4M OB (189 ± 30%) and CC (144 ± 2%) in *Plp1*-null mice. *Plp1*-eGFP cells were also increased in 4M OB (145 ± 3%) and CC (140 ± 4%) in *Plp1*-null mice. WT and *Plp1*-null samples compared using repeated measures ANOVA. *p<0.01.

DOI: https://doi.org/10.7554/eLife.34783.002

The following figure supplements are available for figure 1:

**Figure supplement 1.** Semi-automatic quantification of the density of labeled oligodendrocytes in tissue sections implemented in MATLAB.
DOI: https://doi.org/10.7554/eLife.34783.003
**Figure supplement 2.** 6M old *Plp1*-null mice exhibit regional differences in oligodendrocyte density.
DOI: https://doi.org/10.7554/eLife.34783.004
**Figure supplement 3.** Region-specific astrogliosis was observed in white matter of 2M- old *Plp1*-null mice.
DOI: https://doi.org/10.7554/eLife.34783.005

In the adult brain, there are two sources of oligodendrocytes: local oligodendrocyte progenitor cells (OPCs) and progenitors in the germinal zone of the SVZ (*Figure 2A*) (*Tepavčević et al., 2011*). In 2M mice, neither PLP1 protein nor *Plp1*-eGFP were expressed in either NG2+ OPCs or Sox2 + SVZ cells (*Figure 2—figure supplement 3*). We used EDU to label local proliferation and determine whether local OPCs or SVZ cells had increased proliferation. We observed the number of EDU + cells 2 hr (P18 and 2M) or 4 hr (4M) after injection to label 1 or 2 cell cycles, respectively. In the *Plp1*-null mice only the SVZ and RMS (Sox2+ EDU+ cells) had increased proliferation (*Figure 2F,G*). No local proliferation increase occurred in the OB, CC, M1, or HP (*Figure 2H*). In addition, the density of NG2+ OPCs was unaltered in 4M *Plp1*-null mice, and the percentage of NG2+ OPCs that expressed EDU 2 hr post injection was comparable for WT and *Plp1*-null mice (*Figure 2I*, *Figure 2— figure supplement 4*).

SVZ progenitor cells normally give rise to migratory progenitors that travel within the RMS and differentiate into neurons and oligodendrocytes in the OB (*Tepavčević et al., 2011*). The density of EDU+ neurons (NeuN+) and the percentage of NeuN+ cells that were EDU+ 3 weeks after EDU injection, were not altered in 4M *Plp1*-null mice (*Figure 2—figure supplement 5*). On the other hand, the number of EDU+ oligodendrocytes (SOX10+and *Plp1*-eGFP+) had increased (*Table 2*). These data suggest no gross alteration in neurogenesis in *Plp1*-null mice. However, Pax6+ cells were significantly reduced in the main and accessory olfactory bulb, indicating a specific loss of Pax6

**Table 1.** Abbreviations for brain areas.

| AC | Anterior commissure |
|---|---|
| BS | Brainstem |
| CB | Cerebellum |
| CC | Corpus callosum |
| CX | Cortex |
| EPL | External plexiform layer |
| FI | Fimbria |
| HP | Hippocampus |
| LOT | Lateral olfactory tract |
| M1 | Motor cortex |
| OB | Olfactory bulb |
| PC | Piriform cortex |
| ST | Striatum |
| TH | Thalamus |

DOI: https://doi.org/10.7554/eLife.34783.006

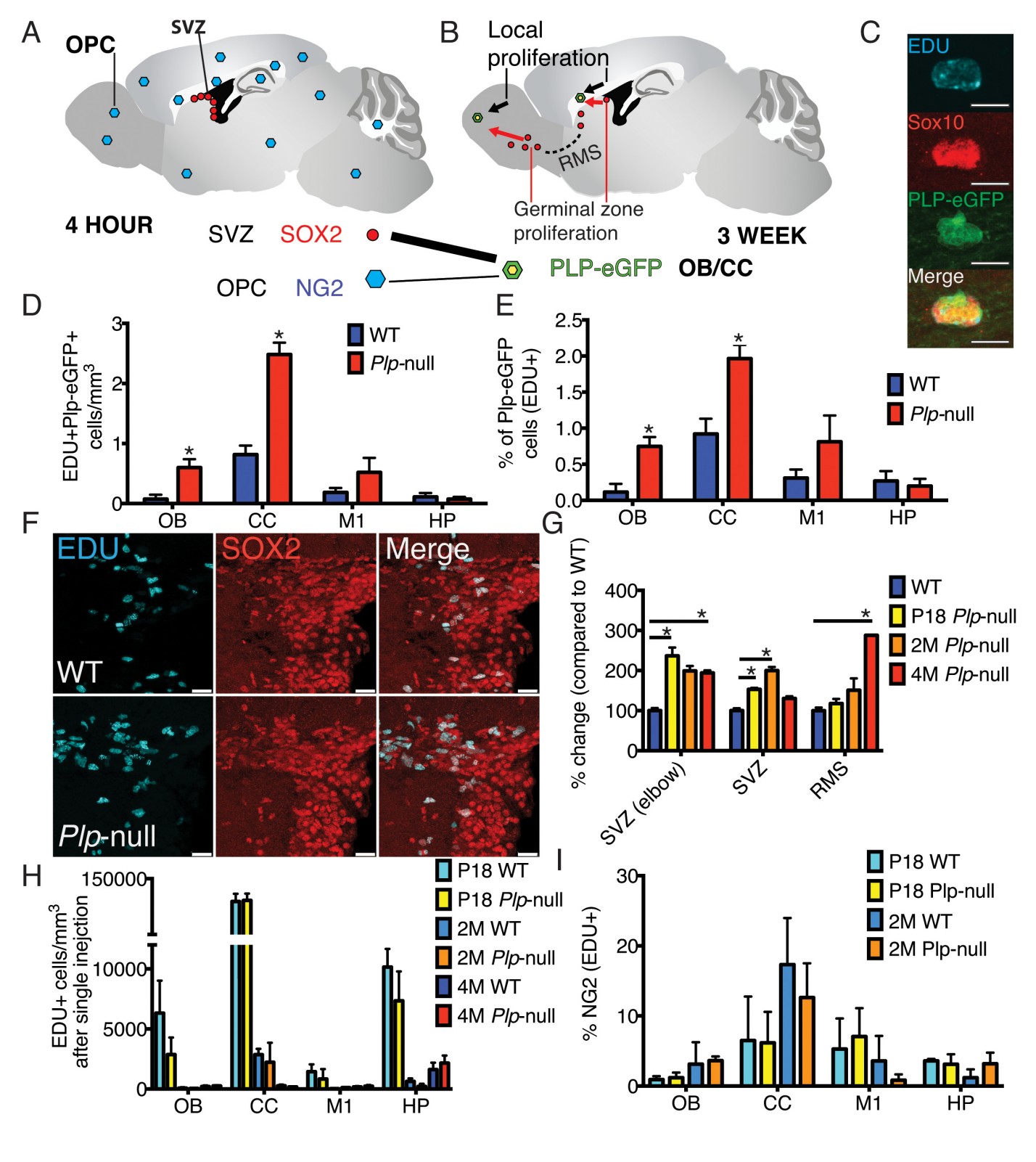

**Figure 2.** New oligodendrocytes in 4M *Plp1*-null mice. (**A**) Location of proliferating NG2+ OPCs throughout the brain, or SOX2+ progenitors in the SVZ 4 hr after a single EdU injection. (**B**) EdU-labeled cells at 4 hr give rise to EdU + *Plp*-eGFP + oligodendrocytes 3 weeks after injection. (**C**) New oligodendrocytes identified using Sox10 and *Plp1*-eGFP. Bar = 10 μm. (**D-E**) Increase in density of EdU + *Plp1*-eGFP + oligodendrocytes (**D**) and percent of *Plp1*-eGFP+ cells expressing EdU (**E**) in *Plp1*-null and WT mice after 3 weeks (*p<0.01, n = 3/genotype). (**F**) Sox2 and EDU cells in the SVZ in *Figure 2 continued on next page*

Figure 2 continued

WT and *Plp1*-null samples. Bar = 25 µm. (G) % change (*Plp1*-null to WT) of SOX2+ progenitors that were EdU+ after a single EdU injection in SVZ or RMS. Compared by repeated measures ANOVA. *p<0.01. (H) No differential density of EdU + cells was observed in the OB, CC, M1 or HP after a single injection. (I) The % of NG2 +cells labeled by single EdU injection was comparable at P18 or 2M.

DOI: https://doi.org/10.7554/eLife.34783.007

The following figure supplements are available for figure 2:

**Figure supplement 1.** No alteration was found in oligodendrocyte cell death assayed with TUNEL in *Plp1*-null mice.

DOI: https://doi.org/10.7554/eLife.34783.008

**Figure supplement 2.** Density of mature oligodendrocytes (Gst-pi+).

DOI: https://doi.org/10.7554/eLife.34783.009

**Figure supplement 3.** PLP1 expression observed by Plp1-eGFP transgene expression (green) and PLP1 staining (blue) indicates that myelinated fibers are located near and possibly within the SVZ (red line).

DOI: https://doi.org/10.7554/eLife.34783.010

**Figure supplement 4.** No alteration in NG2 +OPC density in *Plp1*-null mice.

DOI: https://doi.org/10.7554/eLife.34783.011

**Figure supplement 5.** Neurogenesis in *Plp1* -null mice was unaltered.

DOI: https://doi.org/10.7554/eLife.34783.012

**Figure supplement 6.** Pax6 expression in the RMS.

DOI: https://doi.org/10.7554/eLife.34783.013

+ neurons (*Figure 2—figure supplement 6*). In addition, the Pax6+ cells expressing Sox2, a marker of progenitors was reduced, indicating a reduction in neural progenitors in the RMS (*Figure 2—figure supplement 6*). Therefore, the increase in oligodendrocytes occurs at the expense of the production of Pax6+ neurons.

## Axonal disruption

The SVZ generates oligodendrocytes that contribute to remyelination (*Tepavčević et al., 2011*). We hypothesize that SVZ activation in *Plp1*-null mice is initiated through signals related to myelin disruption and axonal impairment. Optic nerve axons develop spheroids as early as 2M (*Griffiths et al., 1998*). Therefore, we expected to see axonal disruption prior to SVZ activation and the increase in oligodendrocytes. Thy1-YFP transgene was crossed into the *Plp1*-null mouse and YFP+ spheroids were found in axons of projection neurons at 4M (*Figure 3A,B,C*). No loss of layer V Thy1-YFP+ neurons was observed (*Figure 3G*). SMI-32+ spheroids were already visible at 2M in *Plp1*-null mice (*Figure 3E*). Interestingly, in P18 axons, both WT and *Plp1*-null exhibited a wavy phenotype, but no spheroids greater than 10 µm (YFP or SMI-32+) were observed (*Figure 3A*). Thus, SVZ activation and increase in *Plp1*-null oligodendrocytes occurred before observable accumulation of pathologic spheroids.

## *Plp1*-null mice display deficits in conduction velocity in the corpus callosum

Changes in oligodendrogenesis and axonal disruption may lead to changes in axonal conduction velocity (CV). We performed measurements of CV in the CC of WT and *Plp1*-null mice. *Figure 4* shows that the fast transmission N1 peak of the compound action potential (CAP) elicited by electrical stimulation of the CC is absent in 2M and 6M *Plp1*-null mice, but not in P18 *Plp1*-null mice. In addition, the latency to the slower N2 peak is longer (N1 and N2 peaks of the CAP were defined as in *Crawford et al. (2009)*). These data indicate that although the conduction velocity is initially normal, it is eventually slower in the *Plp1*-null mouse, which is relevant for interhemispheric coordination of neural activity, and is likely to alter behaviors such as motor coordination.

## *Plp1*-null mice were tested for performance in a battery of behavioral tasks

Adult *Plp1*-null mice are behaviorally grossly normal (*Klugmann et al., 1997*). However, the local increase in oligodendrocytes in OB and CC starting at day 18 and subsequent changes in axonal swelling and CV at 2M suggest that circuit function and behavior are altered in *Plp1*-null mice 2M or older. We tested WT and *Plp1*-null mice with a battery of behaviors to determine whether a select

**Table 2.** Accumulation of EDU+ oligodendrocytes in *Plp1*-null mice.
WT and *Plp1*-null mice were compared with a repeated measures ANOVA, n = 3 per genotype. Significant differences (p<0.05) are shown in red font.

| | | Region | Mean(SEM), p value | | |
|---|---|---|---|---|---|
| **3 weeks** | ***Plp1*-eGFP + EDU +** | | **WT** | **Null** | **p value** |
| | (cells/mm$^3$) | OB | 0.07 (0.07) | 0.60 (0.14) | 0.02 |
| | | CC | 0.81 (0.15) | 2.5 (0.20) | <0.01 |
| | | M1 | 0.19 (0.07) | 0.52 (0.24) | 0.11 |
| | | HP | 0.11 (0.06) | 0.07 (0.04) | 0.85 |
| 3 weeks | % *Plp1*-eGFP (EDU+) | | | | |
| | | OB | 0.11 (0.11) | 0.75 (0.13) | 0.11 |
| | | CC | 0.92 (0.21) | 2.00 (0.20) | <0.01 |
| | | M1 | 0.31 (0.12) | 0.81 (0.36) | 0.27 |
| | | HP | 0.23 (0.14) | 0.20 (0.10) | 0.99 |
| 3 weeks | SOX10+ EDU+ | | | | |
| | (cells/mm$^3$) | OB | 0.35 (0.07) | 1.1 (0.02) | 0.04 |
| | | CC | 0.81 (0.15) | 2.5 (0.20) | <0.01 |
| | | M1 | 0.46 (0.15) | 1.2 (0.07) | 0.07 |
| | | HP | 0.81 (0.13) | 1.4 (0.43) | 0.13 |
| 3 weeks | % SOX10 (EDU+) | | | | |
| | | OB | 0.95 (0.25) | 2.1 (0.11) | 0.19 |
| | | CC | 0.90 (0.25) | 1.9 (0.15) | 0.36 |
| | | M1 | 1.8 (0.36) | 3.0 (0.8) | 0.17 |
| | | HP | 1.6 (0.57) | 3.0 (0.24) | 0.09 |
| 3 weeks | % EDU (SOX10+) | | | | |
| | | OB | 1.6 (0.45) | 6.6 (1.0) | 0.86 |
| | | CC | 70 (2.8) | 94 (4.5) | <0.01 |
| | | M1 | 91 (4.8) | 89 (4.7) | 0.99 |
| | | HP | 17 (5.2) | 43 (4.7) | <0.01 |

DOI: https://doi.org/10.7554/eLife.34783.014

subset are altered. We focused on testing motor coordination, problem solving and sensory processing because these behaviors are known to be altered at early stages in demyelinating disorders (*Amboni et al., 2013*; *Ruan et al., 2012*).

## *Plp1*-null mice display subtle motor deficits

Young adult *Plp1*-null mice did not display deficits on the rotarod, a classic test of gross motor function (*Figure 5A*), consistent with previous studies (*Griffiths et al., 1998*). To examine finer motor coordination, we analyzed the gait and swimming patterns. *Plp1*-null mice did not differ in gait from WT at 18 days of age. We performed gait measurements on four 18-day-old mice, two *Plp1*-null males and two wild-type males. The mean stride length for *Plp1*-null animals was 4 cm with a standard deviation of 0.15 cm. Mean stride length for wild-type animals was 3.92 cm with a standard deviation of 0.3 cm. Front-hind paw offset for *Plp1*-null animals was 0.2 cm with a standard deviation of 0.08 cm. Wild-type animals front-hind paw offset was 0.21 cm with a standard deviation of 0.2 cm. These values were not significantly different (Wilcoxon rank-sum p=0.67 for stride length and p=0.78 for paw offset). However, 3M- and 9M-old *Plp1*-null mice overstepped forelimbs with respect to hindlimbs and displayed longer stride length (*Figure 5B–D*).

We then tested coordinated swimming (*Brooks and Dunnett, 2009*). Swim distance was significantly reduced in 9M *Plp1*-null mice (*Figure 5F*) and mean velocity was reduced in 3M and 9M *Plp1*-

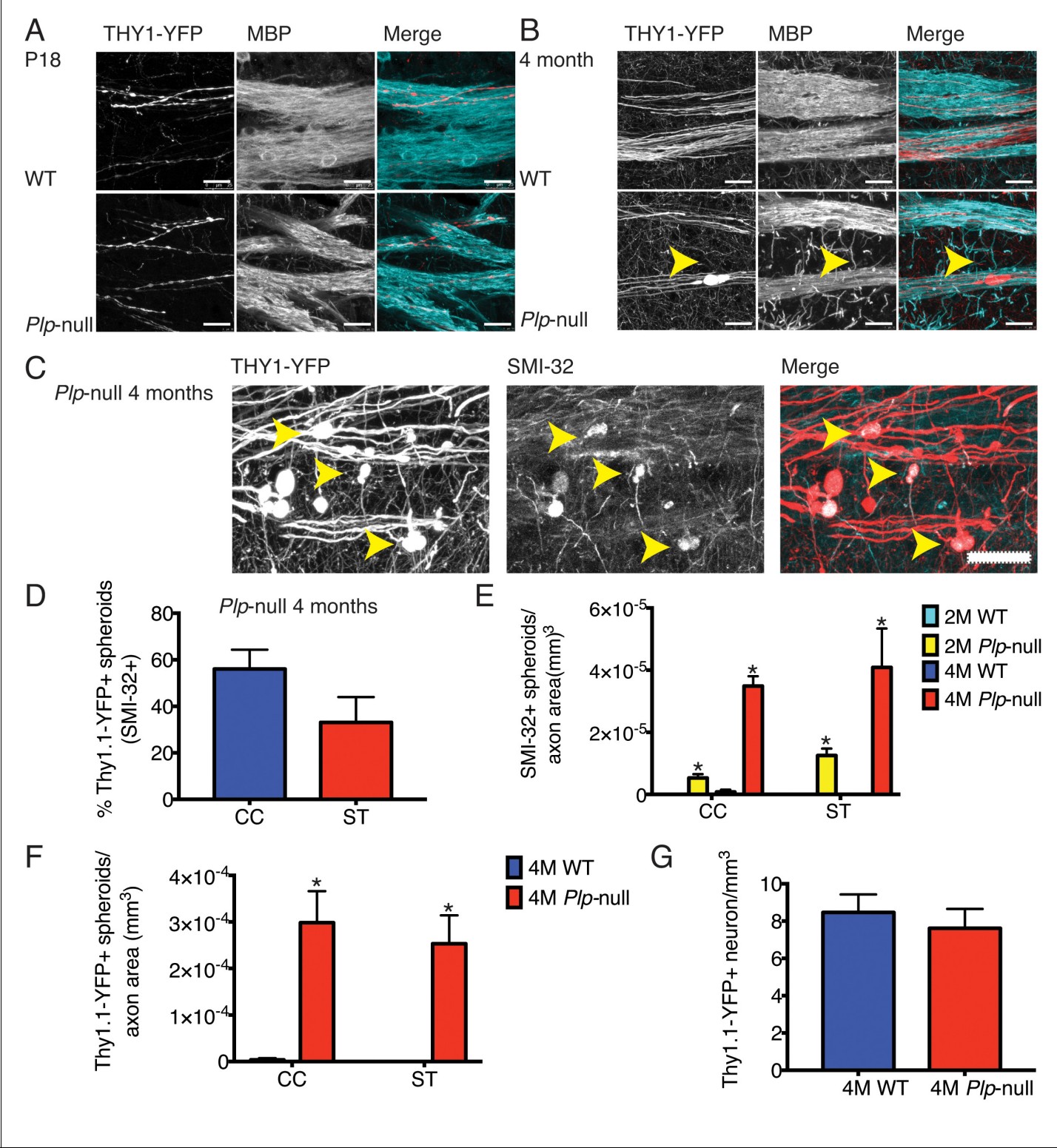

**Figure 3.** Cortical axon disruption in *Plp1*-null mice. (**A**) Thy1-YFP-positive axons were myelinated in P18 striatum and were surrounded by myelin basic protein (MBP) +processes. At P18, the developing Thy1-YFP axons had varicosities in both WT and *Plp1*-null mice. Scale bar = 20 µm. (**B**) At 4 months, a significant number of spheroids were visible in *Plp1*-null mice in the striatum. Scale bar = 20 µm. (**C**) SMI-32 was observed in Thy1-YFP+ spheroids in 4M *Plp1*-null mice. Yellow arrowhead indicates location of spheroid. Scale bar = 20 µm. (**D**) Percentage of Thy1-YFP+ spheroids that were SMI-32+ in 4M *Plp1*-null mice in CC (56 ± 8%) and striatum (ST) (33 ± 11%). (**E**) Density of SMI-32+ spheroids increased in 2M (0 ± 0 versus 5.0 × 10^{−5} ± 1.0 × 10^{−5})

*Figure 3 continued on next page*

*Figure 3 continued*

and 4M *Plp1*-null mice ($7.4 \times 10^{-7} \pm 7.4 \times 10^{-7}$ versus $3.0 \times 10^{-4} \pm 3.0 \times 10^{-6}$). (F) A significant increase in the number of spheroids per axon area was observed in 4M Thy1-YFP + *Plp1* null mice. (G) No change in the density of Thy1-YFP+ layer V neurons was observed. n = 3 mice/genotype. WT and *Plp1*-null were compared with a repeated measures ANOVA. *p<0.01.

DOI: https://doi.org/10.7554/eLife.34783.015

null mice (*Figure 5G*, *Figure 5—videos 1* and *2*). We used a coding scheme to evaluate swimming pattern (score of 0 to 3, see Materials and methods). *Plp1*-null mice never had a coordinated swimming pattern, that is, the highest score (*Mighdoll et al., 2015*) was never observed in 3M or 9M *Plp1*-null mice (*Figure 5E*, *Figure 5—figure supplement 1*).

The altered gait pattern and lack of coordinated swimming indicate a deficit in motor coordination in young *Plp1*-null mice preceding gross motor deficits at 16M (*Griffiths et al., 1998*). The deficits in motor coordination suggest a disruption of the integration of proprioceptive cues necessary to produce a coordinated motor pattern. These subtle motor deficits did not interfere with general locomotion in other behavioral assessments (*Table 3*).

## Cognitive function is impaired in *Plp1*-null mice

Gait and cognition are highly related in humans, and deficits in gait precede declines in cognitive function in neurological disease (*Amboni et al., 2013*). We hypothesized that these mice could have a broader deficit in information processing leading to cognitive deficits. In order to evaluate cognition/executive function, we tested *Plp1*-null mice in the Puzzle Box, a comprehensive test of learning, memory and problem-solving (*Ben Abdallah et al., 2011*). In the Puzzle Box, mice must reach a dark goal box to escape a brightly lit arena, and access to the goal box is made progressively more difficult (*Figure 6A*). The time of entry to the dark goal box in each stage of this task depends on problem solving of novel entry constraints within each stage using skills learned in previous stages. Progressive difficulty is reflected by longer time to enter in the latter stages (*Figure 6B,C*). *Plp1*-null

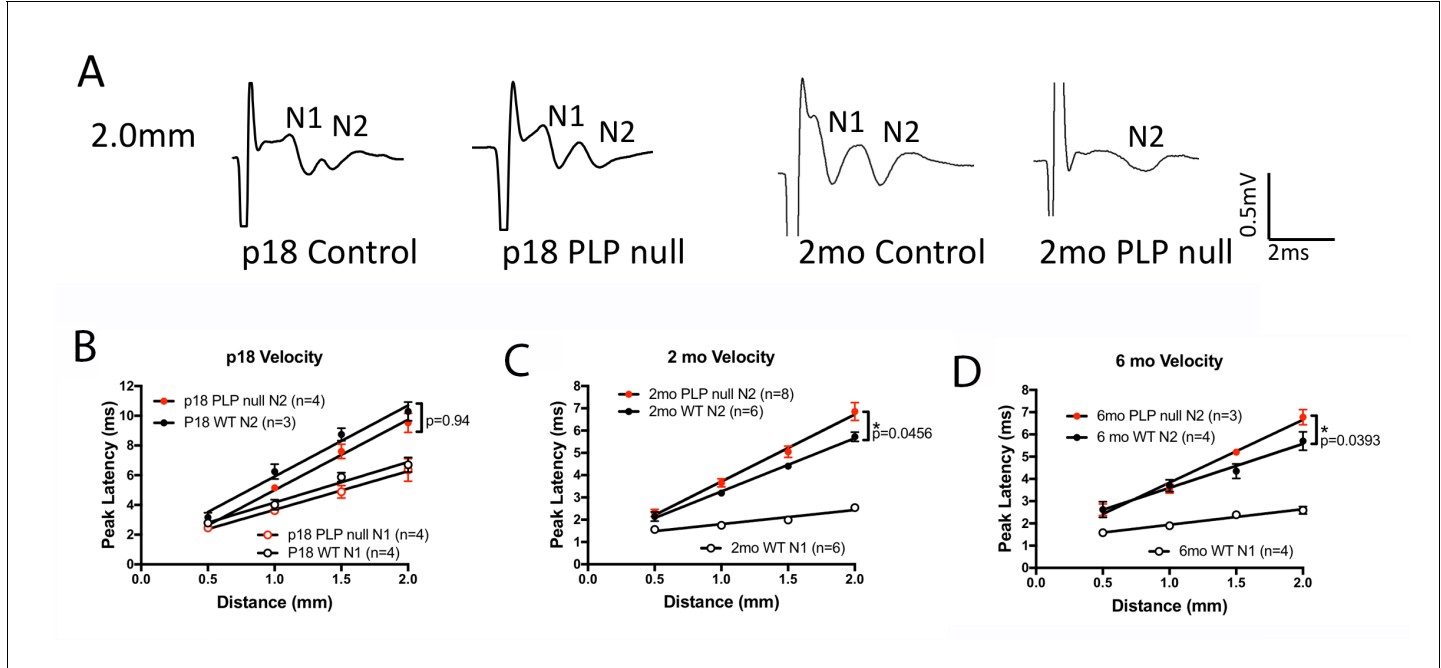

**Figure 4.** The compound action potential (CAP) for the corpus callosum differs between *Plp1*-null mice and controls. (A) Representative traces of callosal CAP recordings elicited by electrical stimulation 2.0 mm away. Fast (N1) and slow (N2) peaks are labeled. The N1 peak is missing for the 2M *Plp1*-null trace. (B-D) Peak latency vs. distance data for 18D (B), 2 M (C) and 6 M (D) mice. Conduction velocity: N1 ~1.5 m/s. p values are shown for analysis of covariance.

DOI: https://doi.org/10.7554/eLife.34783.016

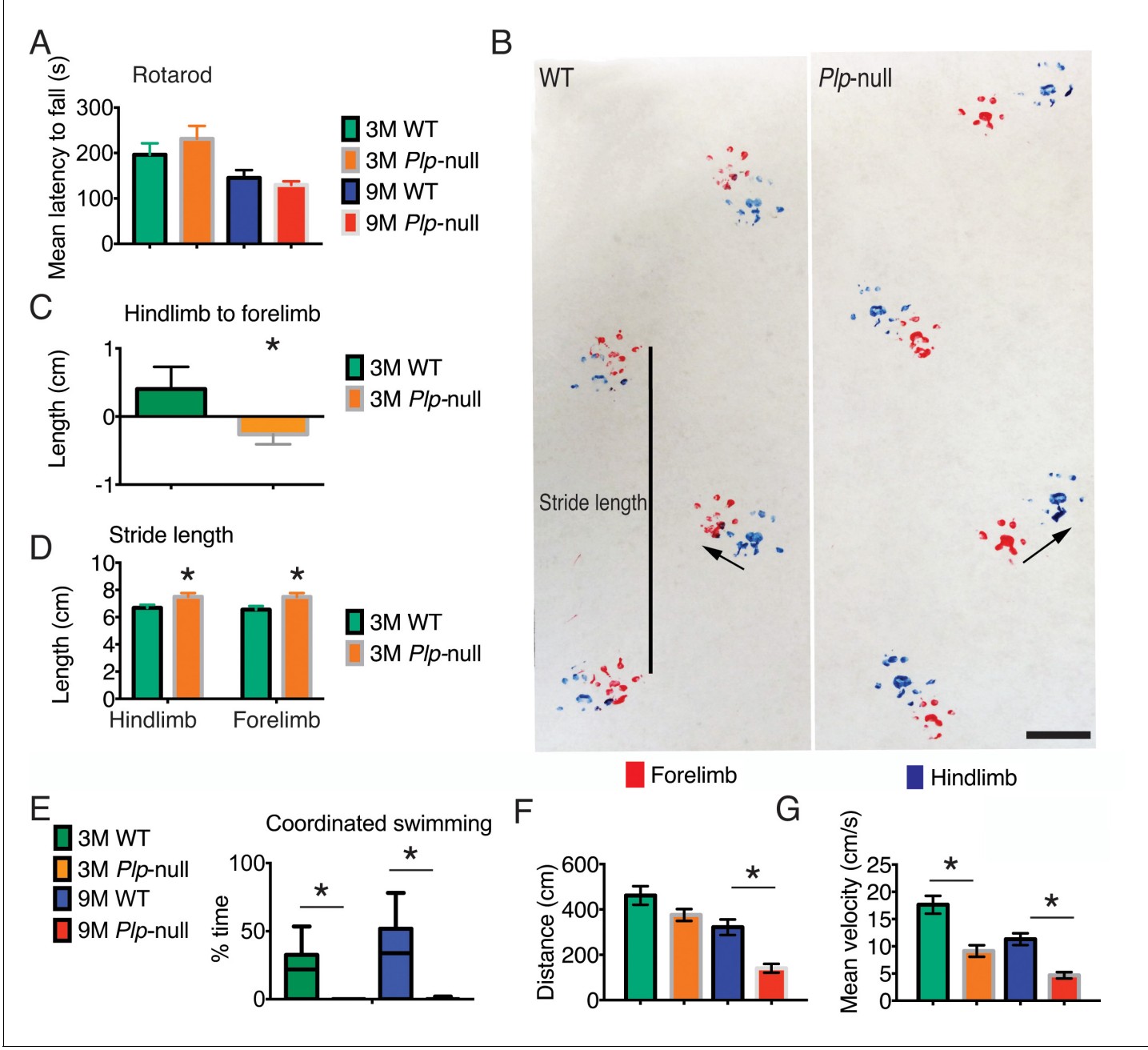

**Figure 5.** Motor deficits in *Plp1*-null mice. (**A**) Rotarod performance was not altered in 3 and 9M *Plp1*-null mice. Mean latency to fall (average of three trials, two-way ANOVA, p=0.64, n = 10/ genotype). (**B**) Gait pattern of 3M *Plp1*-null and WT siblings. 3M *Plp1*-null mice overstep their hind limbs as indicated by the arrows. Bar = 1 cm. (**C**) The directionality from hindlimb to forelimb was reversed in 3M *Plp1*-null mice (0.4 ± 0.1 vs −0.3 ± 0.03 cm; unpaired t-test, n = 10/ genotype, *p<0.01). (**D**) *Plp1*-null mice display a greater stride length (forelimb: 7.2 ± 0.2 vs 6.3 ± 0.1 cm; hindlimb: 7.0 ± 0.2 vs 6.3 ± 0.1 cm; unpaired t-test, n = 10/ genotype, *p<0.01). (**E**) Coordinated swimming pattern was never observed in *Plp1*-null mice. Swimming patterns were quantified (Materials and methods) and *Plp1*-null mice scored 0 (3M: 26 ± 7.6 vs 0 ± 0, 9M: 35 ± 6.2 vs 0 ± 0; two-way ANOVA, n = 10/genotype, *p<0.01). (**F-G**) 9M *Plp1*-null mice (G: distance traveled = 140 ± 19 cm; F: mean velocity = 4.7 ± 0.6) performed worse than WT (G: distance traveled = 321 ± 34 cm; F: mean velocity = 11 ± 1.1; two-way ANOVA, n = 10 mice/genotype, *p<0.01). (**G**) 3M *Plp1*-null displayed slower mean velocity (F: 9.1 ± 1.1 vs 18 ± 1.6 cm/s; two-way ANOVA, n = 10 mice/genotype, *p<0.01). Means ± s.e.m (**A,C,D,F,G**) or whisker plot showing min-max (**E**).

DOI: https://doi.org/10.7554/eLife.34783.017

The following video and figure supplement are available for figure 5:

**Figure supplement 1.** Swim scale evaluation of coordinated swimming.

DOI: https://doi.org/10.7554/eLife.34783.018

**Figure 5—video 1.** Movie showing the swimming pattern for a WT mouse.

*Figure 5 continued on next page*

*Figure 5 continued*

DOI: https://doi.org/10.7554/eLife.34783.019

**Figure 5—video 2.** Movie showing the swimming pattern for a *Plp1*-null mouse.

DOI: https://doi.org/10.7554/eLife.34783.020

mice (3M and 9M) took longer to enter in stage three when the entry-way was filled with saw dust (***Figure 6B,C***), and 9M-old *Plp1*-null mice took longer to enter in the fourth stage when the entry

**Table 3.** Distance traveled and velocity did not differ between WT and *Plp1*-null mice in a subset of behavioral tasks. WT and *Plp1*-null mice were compared with a repeated measures ANOVA, n = 10 per genotype. There were no significant differences (p>0.05).

| Behavioral performance | 3 months | | | 9 months | | |
|---|---|---|---|---|---|---|
| | WT (n = 10) Mean(SEM) | Null (n = 10) Mean(SEM) | p value | WT (n = 10) Mean(SEM) | Null (n = 10) Mean(SEM) | p value |
| Open field (duration in center (s)) | 217.0 (15.52) | 203.71 (0.336) | 0.88 | **238.5 (25.14)** | **158.6 (22.99)** | **0.02** |
| Zero maze (duration open arms (s)) | **92.65 (17.03)** | **175.7 (21.24)** | **0.03** | 126.5 (27.41) | 117.2 (26.09) | 0.95 |
| Y maze (% successful alternations) | 61.48 (2.484) | 61.56 (2.220) | 0.99 | 57.33 (2.888) | 57.32 (3.726) | 0.99 |
| Y maze (arm entries) | 32.18 (2.529) | 35.91 (2.387) | 0.60 | 32.20 (3.608) | 29.80 (3.252) | 0.82 |
| Marble burying (marbles buried) | **4.800 (1.052)** | **1.600 (0.400)** | **<0.01** | **3.900 (0.862)** | **0.200 (0.133)** | **<0.01** |
| Marble burying (time digging (s)) | 8.428 (2.245) | 2.994 (0.666) | 0.25 | **13.24 (3.767)** | **4.294 (2.390)** | **0.04** |
| Locomotion | | | | | | |
| Rotarod (mean latency to fall (s)) | 196.2 (25.08) | 231.2 (28.41) | 0.88 | 145.1 (17.38) | 129.7 (7.948) | 0.94 |
| Distance traveled (cm) | | | | | | |
| Open field | 5255 (466.3) | 4589 (229.8) | 0.39 | 4510 (444.4) | 5205 (322.4) | 0.36 |
| Zero maze | 9532 (3170) | 10831 (2210) | 0.92 | 6148 (1708) | 7030 (1694) | 0.96 |
| Y maze | 27693 (1346) | 15484 (11985) | 0.68 | 3931 (734.5) | 3812 (324.3) | 0.99 |
| Marble burying | 3729 (379.1) | 2931 (361.1) | 0.16 | 5591 (204.0) | 5566 (289.0) | 0.99 |
| Habituation | | | | 845.2 (78.3) | 1041 (56.31) | 0.06 |
| Velocity (cm/s) | | | | | | |
| Open field | 9.551 (0.673) | 9.287 (0.336) | 0.94 | 7.518 (0.741) | 8.667 (0.537) | 0.32 |
| Zero maze | 16.34 (6.224) | 10.25 (2.864) | 0.52 | 11.79 (2.815) | 18.38 (3.695) | 0.46 |
| Y maze | 65.95 (43.50) | 33.15 (24.92) | 0.61 | 8.987 (1.585) | 8.198 (0.885) | 0.99 |
| Marble burying | 6.766 (0.407) | 6.188 (0.595) | 0.61 | 9.323 (0.339) | 9.298 (0.482) | 0.99 |
| Habituation | | | | 7.859 (0.619) | 6.986 (1.028) | 0.47 |
| Distance traveled (cm) | | | | | | |
| Open field | 5255 (466.3) | 4589 (229.8) | 0.39 | 4510 (444.4) | 5205 (322.4) | 0.36 |
| Zero maze | 9532 (3170) | 10831 (2210) | 0.92 | 6148 (1708) | 7030 (1694) | 0.96 |
| Y maze | 27693 (1346) | 15484 (11985) | 0.68 | 3931 (734.5) | 3812 (324.3) | 0.99 |
| Marble burying | 3729 (379.1) | 2931 (361.1) | 0.16 | 5591 (204.0) | 5566 (289.0) | 0.99 |
| Habituation | | | | 845.2 (78.3) | 1041 (56.31) | 0.06 |
| Velocity (cm/s) | | | | | | |
| Open field | 9.551 (0.673) | 9.287 (0.336) | 0.94 | 7.518 (0.741) | 8.667 (0.537) | 0.32 |
| Zero maze | 16.34 (6.224) | 10.25 (2.864) | 0.52 | 11.79 (2.815) | 18.38 (3.695) | 0.46 |
| Y maze | 65.95 (43.50) | 33.15 (24.92) | 0.61 | 8.987 (1.585) | 8.198 (0.885) | 0.99 |
| Marble burying | 6.766 (0.407) | 6.188 (0.595) | 0.61 | 9.323 (0.339) | 9.298 (0.482) | 0.99 |
| Habituation | | | | 7.859 (0.619) | 6.986 (1.028) | 0.47 |

DOI: https://doi.org/10.7554/eLife.34783.021

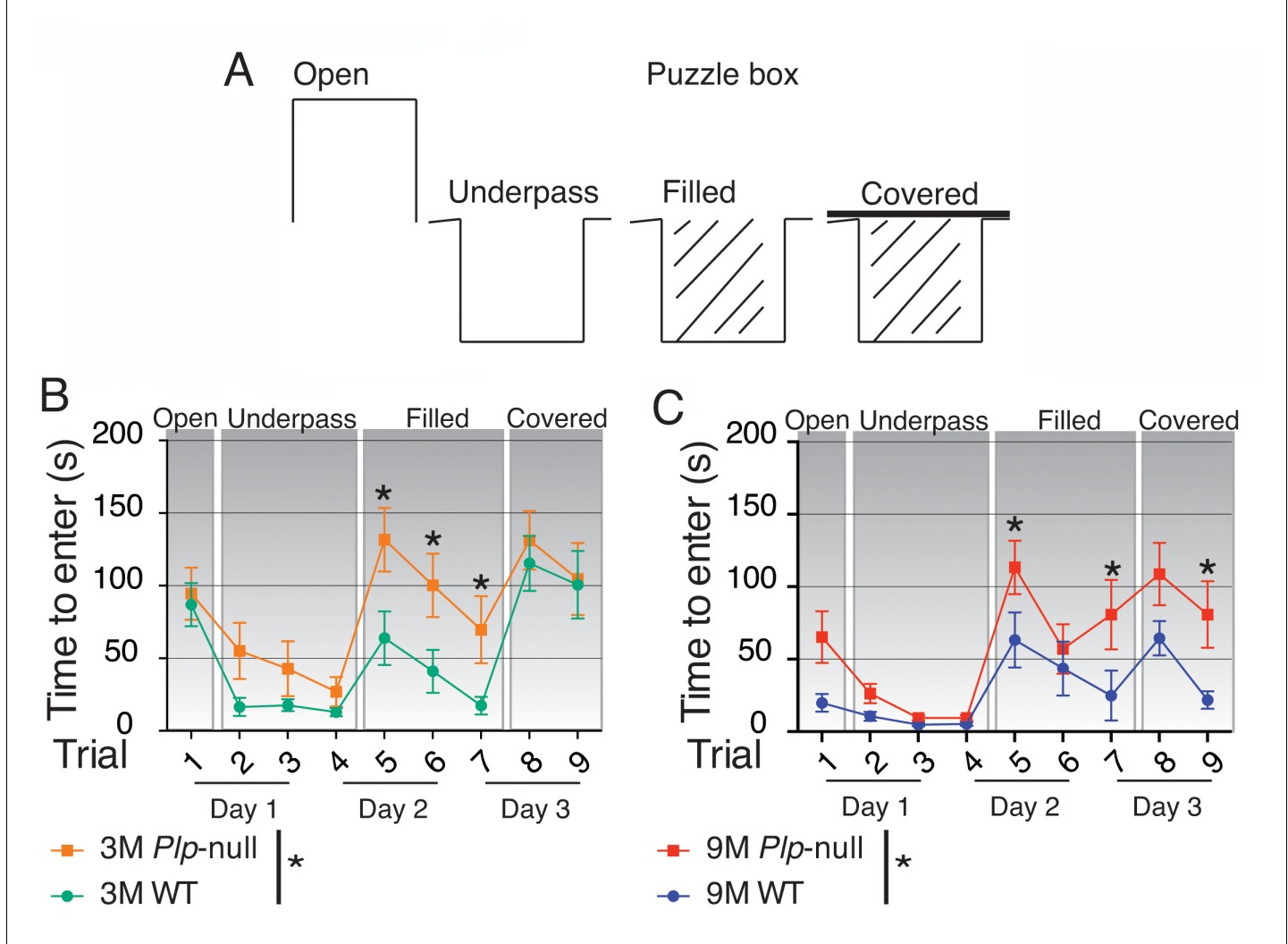

**Figure 6.** Cognitive function is impaired in *Plp1*-null mice. (A) Diagram of entry conditions in the Puzzle Box task. (B) Time to enter the goal box differed significantly between 3M WT and *Plp1*-null mice (F(1,161) =12.2, ANOVA, n = 10/genotype, p<0.01). 3M WT and *Plp1*-null differed in trials 5, 6, and 7 (trial 5: p<0.01, trial 6: p<0.01, trial 7: p=0.03, Sidak's multicomparison, n = 10/genotype). (C) Time to enter differed significantly overall between WT and *Plp1*-null at 9M (F(1,170) =22.7, ANOVA, n = 10/genotype, p<0.01). 9M WT and *Plp1*-null differed in trials 5, 7, and 9 (trial 5: p<0.01, trial 7: p<0.01, trial 9: p=0.03, Sidak's multicomparison, n = 10/genotype).

DOI: https://doi.org/10.7554/eLife.34783.022

The following figure supplements are available for figure 6:

**Figure supplement 1.** *Plp1*-null mice exhibit no alteration in digging behavior in the Puzzle Box task.

DOI: https://doi.org/10.7554/eLife.34783.023

**Figure supplement 2.** The percent successful alternation patterns in Y maze was similar in WT and *Plp1*-null mice (two-way ANOVA, p=0.99, n = 10/genotype).

DOI: https://doi.org/10.7554/eLife.34783.024

**Figure supplement 3.** Marble burying task.

DOI: https://doi.org/10.7554/eLife.34783.025

was covered (*Figure 6C*). No alteration in the digging behavior necessary to dig through the underpass filled with sawdust was noted (*Figure 6—figure supplement 1A,B*). Thus, cognitive impairments were more noticeable after increasing task difficulty. To assess differences in working memory we used the Y maze where differences result in changes in alternation of entry to a novel arm. We observed no deficit in spatial working memory in this behavioral task, as 4M and 9M *Plp1*-null mice performed similarly to WT in the Y maze (*Figure 6—figure supplement 2*, *Table 4*).

**Table 4.** Behavioral performance in the Y maze was similar in WT and *Plp1*-null mice (two-way ANOVA, p=0.99, n = 10/ genotype, p>0.05).

| | 3 month | | | 9 month | | |
|---|---|---|---|---|---|---|
| **Behavioral performance** | **WT (n = 10) Mean(SEM)** | **Null (n = 10) Mean(SEM)** | **p value** | **WT (n = 10) Mean(SEM)** | **Null (n = 10) Mean(SEM)** | **p value** |
| Y maze (% successful alternations) | 61.48 (2.484) | 61.56 (2.220) | 0.99 | 57.33 (2.888) | 57.32 (3.726) | 0.99 |
| Y maze (arm entries) | 32.18 (2.529) | 35.91 (2.387) | 0.60 | 32.20 (3.608) | 29.80 (3.252) | 0.82 |

DOI: https://doi.org/10.7554/eLife.34783.026

## Repetitive perseverative behavior is impaired in *Plp1*-null mice

The *Plp1*-null mice were tested in the marble burying task, which is used to measure repetitive perseverative behavior (*Thomas et al., 2009*). Mice normally exhibit burying behavior defined as displacement of bedding material using the snout and forepaws to cover an object. *Plp1*-null mice displayed a striking behavioral deficit in this task at 3 and 9M (*Figure 6—figure supplement 3*). The marble burying deficit could result from impaired motor coordination necessary to dig. However, digging was not altered in the puzzle box (*Figure 6—figure supplement 1A,B*) and was not significantly reduced in the marble burying task in 3M *Plp1*-null mice (*Figure 6—figure supplement 3C*), suggesting that the perseverative burying deficit occurred prior to the reduced digging behavior. On the other hand, the amount of time spent digging was significantly reduced in 9-month-old *Plp1*-null mice, demonstrating a progressive deficit in burying is associated with a decreased tendency to dig, possibly due to progressive motor impairment (*Figure 6—figure supplement 3C*).

## Tests of anxiety-like behavior are inconclusive for *Plp1*-null mice

The tests of anxiety-like behavior were somewhat inconsistent: increased time in open arms of the zero maze in 3M mice indicated reduced anxiety-like behavior; however, 9M *Plp1*-null mice exhibited increased anxiety-like behavior in the open field (*Table 5*), and, although this was not a statistically significant effect, *Plp1*-null mice exhibited an increased latency to explore the puzzle box entry (*Figure 6—figure supplement 1C,D*). Paradoxical results in the evaluation of anxiety-like behavior have been previously observed in mice overexpressing PLP1, indicating myelin impairments could disrupt emotional regulation (*Tanaka et al., 2009*; *Edgar and Sibille, 2012*).

## Odor perception is impaired in *Plp1*-null mice

There is a remarkable link between olfaction and neurological disorders (*Ruan et al., 2012*). We assessed olfactory function through the go-no-go task where a thirsty mouse licks for a water reward in response to a rewarded odor (*Li et al., 2015*). *Plp1*-null mice were able to distinguish between the odorants (*Figure 7A*), and when they were retested 5 months later (at 8M) they performed as well as at 3M (*Figure 7A*), supporting the conclusion that associative learning of easily differentiated stimuli was normal. We then evaluated odor investigation using the habituation/dishabituation paradigm (*Yang and Crawley, 2009*). The mice investigated synthetic odors less than social odors (*Figure 7B,C*). The investigation of synthetic odors was significantly reduced in 9M *Plp1*-null mice compared to WT (*Figure 7B*). By contrast, novel social odors resulted in a higher dishabituation

**Table 5.** Behavioral characterization of anxiety-like behavior in *Plp1*-null mice.
WT and *Plp1*-null mice were compared with a repeated measures ANOVA, n = 10 per genotype. Significant differences (p<0.05) are shown in red font.

| | 3 months | | | 9 months | | |
|---|---|---|---|---|---|---|
| **Behavioral performance** | **WT (n = 10) Mean(SEM)** | **Null (n = 10) Mean(SEM)** | **p value** | **WT (n = 10) Mean(SEM)** | **Null (n = 10) Mean(SEM)** | **p value** |
| Open field (duration in center (s)) | 217.0 (15.52) | 203.71 (0.336) | 0.88 | **238.5 (25.14)** | **158.6 (22.99)** | **0.02** |
| Zero maze (duration open arms (s)) | **92.65 (17.03)** | **175.7 (21.24)** | **0.03** | 126.5 (27.41) | 117.2 (26.09) | 0.95 |

DOI: https://doi.org/10.7554/eLife.34783.027

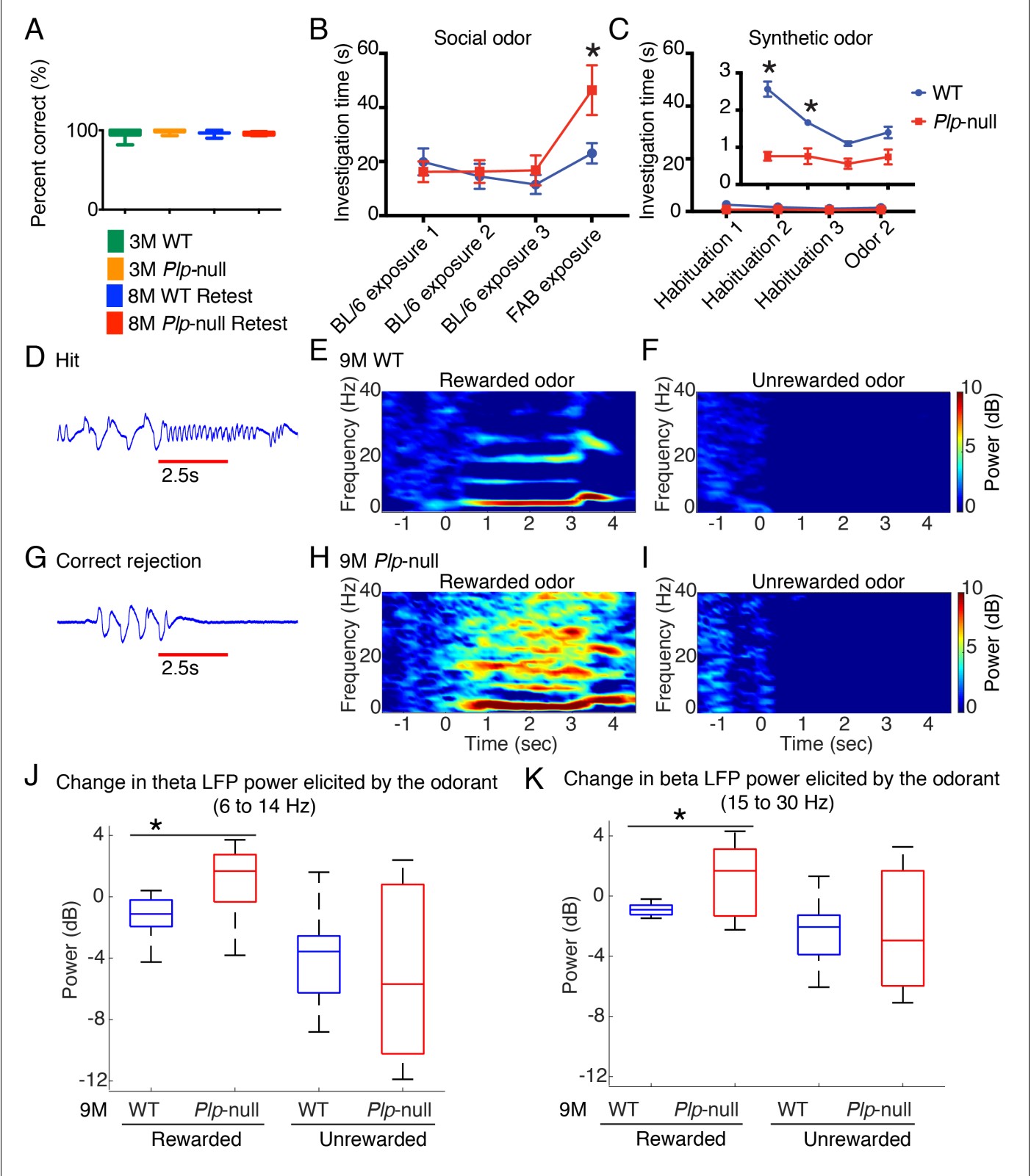

**Figure 7.** Olfactory function. (**A**) *Plp1*-null mice (3M and 8M) performed similarly to WT in go-no-go task (p=0.69, one-way ANOVA, n = 10/genotype). (**B**) 9M *Plp1*-null mice exhibit increased exploration of novel social odor (46 ± 9 vs 23 ± 4 s; 2-way ANOVA, n = 10/genotype, *p<0.01). (**C**) Synthetic odor investigation was reduced in *Plp1*-null mice (F(***Browne et al., 2014***; ***Almeida and Lyons, 2017***) = 30.9; two-way ANOVA, n = 10/genotype, *p<0.01). (**D**) Representative extracellular voltage trace recorded from piriform cortex when the animal is responding correctly to rewarded odor (hit).
*Figure 7 continued on next page*

*Figure 7 continued*

Red line = odor delivery. (**E-F**) Heat maps displaying the stimulus-induced change in power of oscillations during the rewarded (**E**) and unrewarded trials (**F**) in 9M WT mice. (**G**) Voltage trace recorded from piriform cortex during the unrewarded trial (correct rejection). (**H-I**) Heat maps showing the change in the power of oscillations during rewarded trial (**H**) and unrewarded trial (**I**) in 9M *Plp1*-null mice. (**J**) During rewarded trials, 9M *Plp1*-null mice demonstrated odorant-induced increase of power in theta (6 to 14 Hz) (p<0.01, Mann-Whitney U test, n = 64 LFP recordings/genotype). (**K**) 9M *Plp1*-null mice demonstrated a greater increase in the change in beta power during rewarded trials (15–30 Hz) (p<0.01, Mann-Whitney U test, n = 64 LFP recordings/genotype).

DOI: https://doi.org/10.7554/eLife.34783.028

The following figure supplements are available for figure 7:

**Figure supplement 1.** Piriform cortex modeling shows an increase in LFP power when LOT conduction velocity is decreased.

DOI: https://doi.org/10.7554/eLife.34783.029

**Figure supplement 2.** LFP power in WT and *Plp1*-null mice.

DOI: https://doi.org/10.7554/eLife.34783.030

rebound in *Plp1*-null mice (*Figure 7C*), which suggests the salience of the novel social odor was higher in *Plp1*-null mice.

OB output is carried by myelinated axons that converge to form the lateral olfactory tract; therefore, myelination is important for information transmission from OB to piriform cortex. Mild myelin alterations reduce conduction velocity, which will alter oscillations (*Filley and Fields, 2016*; *Pajevic et al., 2014*; *Almeida and Lyons, 2017*). Computational modeling of piriform cortex LFP oscillations indicated that a decrease in conduction velocity would lead to an increase in oscillatory power (*Figure 7—figure supplement 1*), and indeed, awake behaving recordings demonstrated altered oscillations in *Plp1*-null piriform cortex (*Figure 7D–K*). These mice had increased basal power of oscillation and odorant-induced changes in power within the theta and beta bands in rewarded trials (*Figure 7E,F,H,I,J,K*, *Figure 7—figure supplement 2*).

## Discussion

These are the first studies to demonstrate that mild myelin disruption results in enhanced proliferation in the SVZ generating new oligodendrocytes, accompanied by specific behavioral alterations and changes in neural oscillations in young adulthood (*Figure 8*). We characterize the early cellular response to the absence of the important myelin protein, PLP1. PLP1 is not required for myelination (*Klugmann et al., 1997*). However, PLP1 is necessary for oligodendrocyte-mediated support of axonal integrity, and *Plp1*-null mice develop progressive axonal pathology (*Griffiths et al., 1998*). We demonstrate that these early cellular alterations occur prior to axonal swellings, a late sign of axonal disruption.

SVZ progenitor cells are the likely source of the new oligodendrocytes in the CC and OB, given the early increase in EDU+ cells in the SVZ and the proximity of the SVZ to these regions (*Figure 2*). Unexpectedly, local OPCs did not contribute to the increase in oligodendrocytes in *Plp1*-null mice, as local OPC numbers or proliferative rate did not change (*Figure 2*, *Table 2*). The cellular response is not a response to oligodendrocyte death and appears earlier than measurable signs of axonal disruption, e.g. swellings and reduced conduction velocity. One hypothesis is that the absence of PLP1 during embryonic development alters the proliferative capacity and fate of SVZ cells into adulthood. Alternatively, axonal alterations and their downstream response, which are not observable through these methods, could signal for the enhanced proliferation. This hypothesis seems more likely as increased oligodendrogenesis occurs throughout early adulthood and it is likely these signals persist and perpetuate oligodendrogenesis. Finally, we observed region-specific reactive gliosis in the white matter of the 2M old *Plp1*-null mouse (*Figure 1—figure supplement 3*). Whether the reactive gliosis is caused by Plp1 depletion, or is a reaction to axonal disruption is not known. Interestingly, reactive gliosis is found in the CC of animals treated with cuprizone where it is thought to be mediated by Toll-like receptors (*Esser et al., 2017*).

Mild myelin abnormalities are thought to be a primary cause of behavioral alteration (*Poggi et al., 2016*) and are likely involved in mild cognitive impairment in neurodegenerative disorders (*Bartzokis, 2011*). Mice lacking *Plp1* gene expression displayed decreased conduction velocity in the CC at 2M (*Figure 4*) and specific behavioral alterations by 3M (*Figures 5* and *6*), indicative of

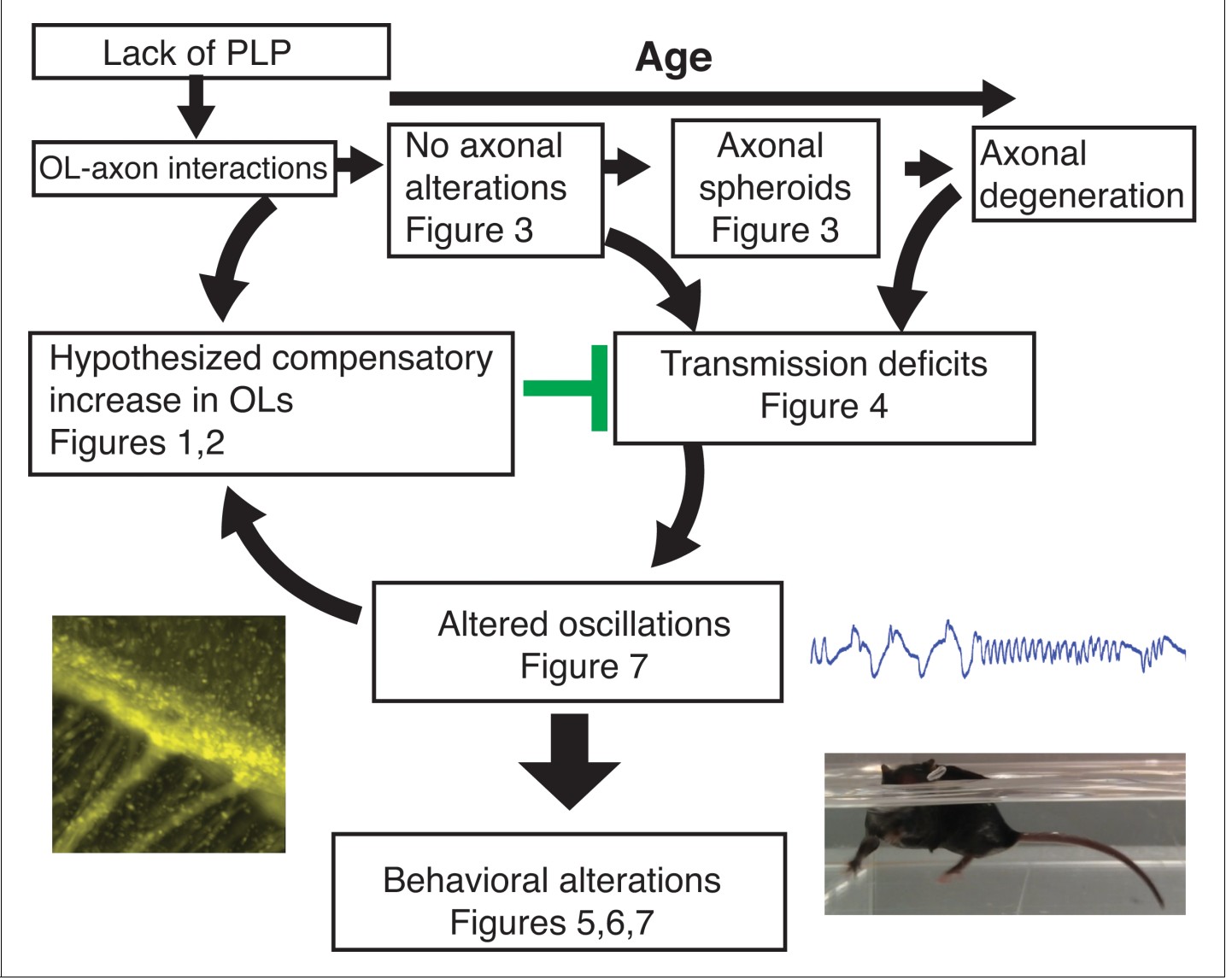

**Figure 8.** The lack of PLP1 dependent signaling led to early (P18) changes in the number of OPCs in the SVZ and CC, followed by formation of axonal spheroids, increased number of OPCs in OB, and glial inflammation at 2M. The cellular changes at 2M were accompanied by loss of the N1 fast conduction velocity peak in the CAP of the CC, which was followed by motor coordination and cognitive deficits at 3M. By 4M, mature oligodendrocytes were increased in the OB and CC. Note: Changes in olfactory function were assayed at 9M and may or may not happen at earlier ages. The images included are *Plp1*-eGFP in the CC (*Figure 1D*) (left), the raw LFP in piriform cortex (*Figure 7D*)(top right) and the *Plp1*-null performing uncoordinated swimming by extending the front paws (*Figure 5—videos 1* and *2*)(bottom right).
DOI: https://doi.org/10.7554/eLife.34783.031

a greater early impact of myelin dysfunction than previously documented (*Griffiths et al., 1998*; *Petit et al., 2014*). *Plp1*-null mice are remarkably normal given the extent of cellular alterations, including an impressive amount of axonal disruption at 2M (*Figure 3*). While *Plp1*-null mice appear grossly normal, and learning and memory are relatively unaltered, cognitive impairments requiring higher-order processing (*Figure 6*) were observed, which became increasingly apparent with task difficulty in the Puzzle Box (*Figure 6*). Overall, the behavioral profile of *Plp1*-null mice indicates that higher order processing is disrupted, supporting the hypothesis that myelin alterations impair timing within neuronal circuits, leading to a deficit in information integration.

The lack of major consequences of the loss of PLP1, a protein that constitutes 20–30% of myelin protein (*Morell et al., 1972*), is surprising. Myelin disruption and associated axonal pathology lead

to conduction velocity delays (*Petit et al., 2014*) (*Figure 4*). It is hypothesized that myelin disruption will lead to alterations in timing of action potential transmission resulting in altered neural oscillations (*Pajevic et al., 2014*). Oscillations are important for transmission of information in neural systems because they provide a syntactical framework for packaging information (*Buzsáki and Watson, 2012*). Here, we find that the myelin/axonal pathology is accompanied by increases in the power of oscillations in the theta and beta frequency (*Figure 7* and *Figure 7—figure supplement 1*). Similar increases in the power of oscillations are observed in MS patients (*Zhou et al., 2014*). Interestingly, optogenetic activation in the beta frequency promotes oligodendrogenesis and adaptive myelination accompanied by behavioral motor improvement (*Gibson et al., 2014*), and recent studies show that optogenetic activation of gamma oscillations attenuates amyloid load in an animal model of Alzheimer's disease (*Iaccarino et al., 2016*), a pathology accompanied by defects in myelination (*Bartzokis, 2011*). We propose that SVZ production of new oligodendrocytes and the accompanying increase in oscillatory power is a protective response reducing behavioral effects of the absence of PLP1 in myelin (*Figure 8*). How a protective increase in oscillatory power is generated is an open question in disorders such as early MS and Alzheimer's. Future studies of compensatory changes in circuit oscillations and SVZ production could yield translationally relevant results for the treatment of progressive behavioral impairment in these disorders.

## Materials and methods

### Animal use

All animals used in this study were treated in accordance with the University of Colorado Animal Care and Use Committee guidelines. We used *Plp1*-eGFP mice that express eGFP under the PLP1 promoter (*Mallon et al., 2002*), *Plp1*-null mice (B6;129-Plp1tm1Kan/J, Jackson Laboratory Stock No: 003255, RRID:IMSR_JAX:003255) (*Griffiths et al., 1998*) and crosses of these two strains. Because the PLP1 gene is found in the X chromosome *Plp1*-null heterozygous females were crossed with wild type (WT) C57BL/6 males (Jackson Laboratories 000664) yielding *Plp1*-null and WT males that were used in all experiments. To study axonal disruption we crossed Plp1-null mice with Thy-1-YFP mice (B6.Cg-Tg(Thy1-YFP)HJrs/J, Jackson Laboratory Stock No 003782, RRID:IMSR_JAX:003782). For all experiments, animals were kept on a reversed light/dark cycle of 14:10 hr with lights on at 10:00 PM. Food (Teklad Global Rodent Diet no 2918; Harlan, Denver, CO) was available ad libitum. All behavior experiments were performed during the dark cycle between the hours of 11 am and 5 pm. All mice were bred in the animal facilities of the University of Colorado Anschutz Medical Campus. Following weaning at 4 weeks, *Plp1*-null male mice and WT male siblings were group housed to prevent any impact of social isolation (*Klugmann et al., 1997*). Whenever possible, behavioral experiments at different ages were performed with the same cohort of animals.

### Preparation and imaging of cleared brain samples

*Plp1*-eGFP mice were used to develop the clearing protocol and measurement techniques (*Mallon et al., 2002*). PACT was performed according to the protocols outlined by the Gradinaru group (*Yang et al., 2014*). *Plp1*-eGFP mice were perfused with 4% paraformaldehyde diluted from 32% paraformaldehyde (Electron Microscopy Sciences, Hatfield, PA, #14714) with 0.1M Phosphate buffered saline (PBS). Brain sections were incubated overnight in 4% acrylamide (Sigma-Aldrich, #A9099) with 0.25% photo-initiator (VA-044, Wako Chemicals USA Inc, Richmond, VA) in 0.1M PBS in 5 mL conical tubes at 4°C. The samples were degassed using a homemade chamber to remove nitrogen and rotated at 37°C for 2 hr. We find that complete degassing is a critical step for sample clarity and antibody labeling. After excess hydrogel was removed by washing three times in 0.1M PBS, tissue sections were placed in 8% SDS in 0.1M PBS in 5-ml conical tubes and rotated at 37°C for multiple days (timing dependent on size and type of tissue) and directly mounted in index matching solution (RIMS), consisting of 40 g of Histodenz (Sigma-Aldrich, #D2158) dissolved in 30 ml of 0.01 M PBS (final concentration of 88% Histodenz w/v). Samples were incubated in RIMS until the brain became transparent.

Cleared brain samples were imaged using cleared tissue digital scanned light-sheet microscopy (C-DSLM), a technique that corrects automatically for spatially heterogeneous refractive index to generate in-focus images of cleared tissue (*Ryan et al., 2017*).

## Immunohistochemical sample preparation

Mice were anaesthetized with avertin or fatal plus before transcardial perfusion. Mice were first perfused with 10 ml of 0.1 M PBS (pH 7.4) followed by 25 ml of 4% paraformaldehyde (PFA) in PBS (pH 7.4) with a flow rate of 12 ml/min. Brains with olfactory bulbs attached were dissected out and postfixed overnight in PFA at 4°C. They were then incubated in cryoprotectant (20% glycerol in 0.1 M Sorensen's buffer, pH 7.6) for 48 hr at 4°C. Free-floating sagittal 30 µm sections were prepared on a cryostat and placed in PBS. For long-term storage, sections were placed in cryostorage solution (30% w/v sucrose, 30% v/v ethylene glycol, 1% polyvinyl-pyrrolidone (PVP-40) in 0.1 M Sorensen's buffer, pH 7.6).

## EDU labeling and quantification of EDU+ cells

For labeling newly generated oligodendrocytes, mice were given i.p. injections of EDU (25 mg/kg) once a day for 3 days and sacrificed 3 weeks after the final injection. For labeling of cells generated within 1–2 cell cycles, mice were sacrificed 2 (P18 and 2 Month mice) or 4 hr (4 Month mice) after a single i.p. injection of EDU (25 mg/kg) was given. EDU click-it kit was used to visualize EDU labeling and was performed according to manufacturer's directions (ThermoFisher, #C10340). EDU+ and SOX2+ cells in the SVZ and RMS were counted by applying surfaces that delineated the cells based on the intensity of the immunostaining with IMARIS (Bitplane, RRID:SCR_007370). EDU+ cells in other regions were quantified using the MATLAB algorithm on tiled sagittal sections.

## Immunohistochemistry

Sections were washed 3x in PBS for 5 min. For Olig2, Sox10, and Sox2, antigen retrieval was performed by microwaving sections in a Pelco Biowave microwave oven 1 × 5 min at 550W in sodium citrate buffer pH 6, followed by washing 1x in PBS for 5 min. They were then blocked for 1–2 hr in blocking solution (5% normal donkey serum, 0.3% Triton-X in PBS). For EDU visualization the EDU click it kit (ThermoFisher #C10340) was performed following blocking step. Sections were incubated in primary antibody diluted in blocking solution overnight at room temperature. Primary antibodies used: goat anti-Sox10 (1:500, Santa Cruz, sc-17342, RRID:AB_2195374), goat anti-Sox2 (1:1000, Santa Cruz, sc-17320, RRID:AB_2286684), Rabbit anti-Olig2 (1:10,000, gift from Dr. Charles Stiles, Harvard University), guinea pig anti-NG2 (1:1000, gift from Dr. William Stallcup, Burnham Institute for Medical Research), mouse anti-NeuN (1:1000, Milipore, MAB377, RRID:AB_2298772), rabbit anti-Pax6 (1:500, Abcam, ab5790, RRID:AB_305110), rat anti-PLP1 (clone AA3)(*Yamamura et al., 1991*). The sections were washed 3x in PBS, followed by a 2-hr incubation in secondary antibody diluted in blocking solution at room temperature. Species-specific donkey secondary antibodies conjugated to Alexa fluorophore 488, 568 or 647 (1:1000, Molecular Probes) were used. The sections were washed 1 × 5 min in PBS followed by a 1 × 5 min in water. They were mounted on charged slides, coverslipped in Fluoromount G, and sealed with clear nail polish. For each antibody, a no primary control was used to verify the specificity of the secondary antibody.

## Labeling of apoptotic cells

Apoptotic cells were labeled using a TUNEL (TdT-mediated dUTP-X nick end labeling) kit (Sigma #12156792910). The TUNEL kit labels DNA fragments with an antibody conjugated to bromodeoxyuridine. Immunohistochemistry was performed as described above to identify the identity of TUNEL + cells that were undergoing apoptosis.

## Digital image acquisition and analysis of tiled sagittal sections

To quantify *Plp1*-eGFP and Olig2+ cells in tiled sagittal sections, tiled z-stacks with a thickness of 10 µm and a step size of 3 µm were taken on a Leica SP5 confocal microscope using a 25x water-immersion objective. Three tiled sections were analyzed per mouse with at least three mice/genotype. The scale of the images is 0.8 µm/pixel. The acquired z-stack was rendered 2D through a maximum intensity projection using the Leica software. Images were analyzed using a MATLAB (RRID:SCR_001622) script run by a user who was blind to the genotype during the entire analysis (*Gould, 2018*). First, background subtraction was performed to render the background more homogenous. For *Plp1*-eGFP+ cells, the background detection eliminated signal from the *Plp1*-eGFP+ oligodendrocyte processes and allowed for better detection of the signal from the cell bodies. To perform

background subtraction the function imopen was used to erode then dilate the image with a disk size determined by the blinded user (average disk size: WT = 8.72 ± 3.61 pixels; *Plp1*-null = 5.26 ± 0.88 pixels). Next, positive pixels were determined using a user-defined threshold value. The threshold value was determined by observing the overlap of the thresholded image and the original image. The average threshold value was 0.14 (±0.02) for WT and 0.15 (±0.03) for *Plp1*-null (in an intensity scale of 0 to 1). Connected objects were identified within the positive pixels using the function bwconncomp. Small objects (<5 pixels) were removed from the analysis since they are too small to be a cell. The percentage of large connected objects over 50 pixels was 2.9% (±1.5) for WT and 0.5% (±0.7) for *Plp1*-null. Therefore, connected objects containing multiple cells comprised only a small percentage of the cells identified. To split the connected objects containing multiple cells, erosion was performed using the function imerode with a disk size of 1 pixel. The erosion splits the connected object into smaller objects that reflect the number of cells within the larger connected object. These smaller objects do not cover the entire cell body; however, the smaller objects created do not have an impact on the overall cell size determined since they comprise a small percentage of the objects quantified. Data was collected in a structural array and exported as a. csv file. To determine the validity this MATLAB algorithm cell counts in a representative area (200 pixels$^2$) from each region examined were manually quantified and then quantified using the MATLAB algorithm. The average percent difference of MATLAB counts relative to manual counts was 4.9% (±4.4%) for WT and 6.4% (±2.5%) for *Plp1*-null.

## Digital image acquisition and analysis of cleared tissue sections

To visualize cleared *Plp1*-eGFP tissue sections, z-stacks were acquired on a homebuilt light-sheet fluorescence microscope (*Ryan et al., 2017*). The light-sheet was generated using a 4x NA 0.14 objective and the Rayleigh length was adjusted using a pinhole to match the sample size. Both 4x NA 0.14 and 10x NA 0.28 air immersion detection objectives were utilized for detection. The axial spacing for each objective was 3.5 µm for the 4x and 1 µm for the 10x objective. This yielded voxel sizes of 1.625 × 1.625 × 3.5 µm for the 4x objective and 0.65 × 0.65 × 1.0 µm for the 10x objective. The laser power was linearly increased through the entire stack through user-defined settings to account for a loss in signal-to-noise as imaging depth increased. At each image plane, an image was captured using both a uniform excitation pattern and sinusoidal excitation pattern with a user defined frequency. These images were used for HiLo image reconstruction to remove effects due to incomplete clearing and scattering (*Mertz and Kim, 2010*). HiLo processed image stacks were then processed such that the mean and standard deviation of each image plane were equalized (*Phair et al., 2004*). Intensity normalized HiLo processed image stacks were then filtered using multiscale adaptive image enhancement (*Zhou et al., 2015*). *Plp1*-eGFP cells were quantified in the final processed image stacks using both the above custom MATLAB code and Vaa3D cell-counting plugin (*Peng et al., 2014*). For MATLAB processing, light-sheet stacks are maximum z projected to match the size of the physical sections.

## Statistical analysis for all imaging methods

Data are represented as mean ±standard error of the mean. For all immunohistochemical studies, an average per mouse was calculated from each set of representative images and at least three mice/genotype were analyzed. Statistical analysis with greater than two comparisons was performed using two-way ANOVA with Sidak's multiple comparison test using Prism statistical software (Graphpad).

## Acute corpus callosum slice preparation

Animals were anesthetized with 3% isoflurane in an $O_2$-enriched chamber. Mice were transcardially perfused with ice-cold (2–5°C) oxygenated (95% $O_2$/5% $CO_2$) artificial cerebral spinal fluid (aCSF) for 2 min prior to decapitation. The brains were then extracted and placed in the same aCSF (in mmol/L: 126 NaCl, 2.5 KCl, 25 NaHCO$_3$, 1.3 NaH$_2$PO$_4$, 2.5 CaCl$_2$, 1.2 MgCl$_2$ and 12 glucose)(*Orfila et al., 2014*). Coronal slices (400 µm thick) were cut with a Vibratome 1200 (Leica) and transferred to a holding chamber containing aCSF at room temperature for at least 1 hr before recording. Slices corresponding approximately to Plates 28–44 in the Franklin and Paxinos atlas (*Paxinos and Franklin, 2012*) were used for recording.

## Corpus callosum electrophysiology

CC conduction velocity was established by changing the distance between stimulating and recording electrodes from 0.5 mm to 2.0 mm, while holding the stimulus intensity constant (*Crawford et al., 2009*). The recording electrode was placed 1 to 1.5 mm away from midline and the stimulating electrode was initially placed 2.0 mm from the recording electrode in the contralateral hemisphere. The stimulating electrode was then moved toward the recording electrode to the closest distance of 0.5 mm in 0.5 mm steps. For analysis of the compound action potential (CAP), the peak latency for N1, where observed, and N2 were measured with four successive data acquisition sweeps per distance and the average of the four sweeps was recorded as the latency for that distance. Linear regression was then performed for each component to yield a slope that is the inverse of the velocity and the slopes were analyzed for statistical comparison using GraphPad.

The amplitude of the N2 component was determined at increasing current steps from 0 to 4.0 mA in 0.25 mA increments. The maximum amplitude of the negative peak was measured in comparison to the preceding positive peak. The CAP amplitude was then graphed versus the stimulus level.

## Behavioral assessment and statistical analysis

In all behavioral assessments, the experimenter is blind to the genotype of the animal. Assessment that required animal tracking was done using a commercial automated animal tracking software (EthoVision XT 8.5 by Noldus, RRID:SCR_000441) and a camera (Panasonic WV-CP284) positioned above the animal. For studies with greater than two comparisons, we utilized a multivariate ANOVA or repeated-measures ANOVA to determine differences between genotypes and different ages or brain regions, respectively. Post-hoc analysis was performed with Sidak's multiple comparison test. A student's T-Test was used to determine differences between two groups. All analyses were run in Prism statistical software (Graphpad, RRID:SCR_002798)

## Rotarod

Rotarod is a standard test of gross motor function (*Brooks and Dunnett, 2009*). Three trials were performed with a 10-min rest period between each trial. The acceleration program went from 3 to 30 rpm over the course of 5 min. The latency to fall was recorded.

## Swimming

Mice were placed in a transparent tank (55cmx33cmx20cm) with the water level at approximately four inches and the mouse was unable to touch the bottom of the tank. Tap water was maintained at 70°F. Mice were not previously exposed to water as this also impacts performance. Mice were recorded from above to assess velocity and distance traveled. To assess swimming pattern the mouse was also recorded using an iPhone 5 positioned at the side of the tank. Mice swam freely for 30 s. A coding scale was used to evaluate the swimming proficiency: 0-floating, 1-forelimbs active, hindlimbs trailing, 2-all four limbs active, 3-tucked forelimbs, hindlimbs engaged.

## Gait analysis

Gait analysis was performed by labeling the footpads with non-toxic tempera paint (*Brooks and Dunnett, 2009*). Footpads were painted with non-toxic tempra paint. Hindlimbs were painted blue and forelimbs were red. The gait pattern was evaluated as the mouse walked down a closed runway. Four to five representative strides were evaluated for each animal using ImageJ to quantify distances.

## Open field

The open-field task examines aspects of anxiety-like behavior (*Prut and Belzung, 2003*). Mice freely explored an open box (44cmx44cmx24cm) while being tracked from above for a 10 min trial. Prior to starting the trial the center and perimeter were delineated using the animal tracking software (Ethovision XT by Noldus). The software was used to determine the amount of time spent in the center versus periphery, as well as the distance travelled.

## Zero maze

The zero maze is a circular runway (diameter = 52 cm) with open and closed portions (*Shepherd et al., 1994*). Mice were tracked for a 10 min trial from above using the animal tracking software (Ethovision XT by Noldus). Prior to starting the trial the open arms and closed arms were delineated by the observer. The software was used to determine the amount of time spent the open and closed arms.

## Y maze

The Y maze has three arms (37 × 7 × 20 cm) shaped like a Y (120° angle). The number of arm entries and which arm was entered was manually recorded while the mouse performed the task. The percent of successful alteration patterns (novel arm to novel arm) was determined (*Hughes, 2004*).

## Marble burying

We followed the procedure of *Thomas et al., 2009*. The number of marbles buried in a 10 min trial was assessed by a blinded observer. The marbles were laid out in a grid of 3 × 3 on top of bedding ~2 inches deep in a small open-field box (28 × 28 × 24 cm).

## Puzzle box

The Puzzle Box was performed as previously described (*Ben Abdallah et al., 2011*). In brief, mice were given 3 min to enter a goal box (15cmx28cmx28cm). If the mouse did not enter the goal box it was encouraged and directed by the experimenter. The covered goal box was adjacent to a well-lit uncovered arena (58cmx28cmx28). The entry of the goal box began as a doorway (*Browne et al., 2014*) then was changed to an underpass (trial 2, 3, 4), the underpass was then filled with sawdust (trial 5, 6, 7), and then the filled underpass was covered with a small rectangular piece of cardboard (trial 8, 9). Each mouse underwent nine trials over a period of 3 days. The first day consisted of trials 1–3, the second day consisted of trials 4–6, and trials 7–9 occurred on the third day.

## Go-no-go odor discrimination task

Odorant-driven active associative learning was used to assess gross odor discrimination (*Li et al., 2015*). Access to water was restricted during the behavioral session; however, if mice failed to receive ~1 ml of water during the behavioral session, additional water was provided in a dish in the bottom of the cage. All mice were weighed daily and maintained at >80% of pre-water restriction weight. No mice dropped below >80% pre-water restriction weight during this study. Mice were trained in an olfactometer chamber to lick in response to the rewarded (S+) odor or refrain from licking in response to the unrewarded (S-) odor, using a go-no-go paradigm (*Li et al., 2015*). For this task, water-deprived mice initiate the trial by placing their head in a sampling chamber for a preafferent period (*Li et al., 2015*). 1–1.5 s after the trial is initiated the odor is delivered. The mouse obtains water-delivery to the rewarded odor when they lick at least once in four 0.5 s segments 0.5–2.5 s after odor onset. There is no penalty for licking prior to odor delivery. The mouse refrains from licking in response to the unrewarded odor due to the substantial effort required for licking the tube. During training mice were presented with 200 trials (block = 20 trials with 10 S + and 10 s− pseudo-randomly selected presentations) and the percent correct for each block was determined. Following completion of criterion (three consecutive blocks ≥ 85%) for two consecutive presentations, training was complete. All mice were trained on a rewarded odor (10% isoamyl acetate (v/v)) vs. unrewarded odor (mineral oil). Mice were then tested on a pair aldehydes (unrewarded: 1% (v/v) heptaldehyde; rewarded: 1% (v/v) heptaldehydes [50%] plus octaldehyde [50%]) at 3 months of age and tested again at 8 months of age with the same odor pairs. The percent correct responses were calculated for the last three trials.

## Olfactory habituation/dishabituation

Synthetic odors were delivered by lightly soaking a piece of filter paper with the odorant. The filter paper was placed inside a cap with two punched holes, to ensure the odor could be detected. The habituated odor was isoamyl acetate (50 μM) and the novel synthetic odor was cineol (50 μM). For social odors, we rubbed the petri dish on the urogenital region of a female BL/6 or FAB background mouse. The mouse was habituated to the petri dish with an unscented piece of filter paper in it prior

to the first trial for a period of 10 min. Habituation to repeated exposure of the same odor was performed by exposing the mouse to 3 2 min trials. On the fourth – 2 min trial, a novel odor was delivered.

## Modeling of piriform cortex neural activity

Simulations were performed in a computational model of the piriform cortex developed on the GENESIS simulator by Michael Vanier (*Vanier, 2001*; *Bower and Beeman, 2007*). A detailed description of the piriform cortex model and parameters can be found elsewhere (*Vanier, 2001*) (reviewed in [*Souza and Antunes, 2007*]). Briefly. the model is composed by conductance-based pyramidal and non-pyramidal cells disposed in a matrix of 16 × 6 positions (96 pyramidal cells) and 15 × 5 positions (75 non-pyramidal cells), respectively. The pyramidal cells are represented by 15 equivalent cylinders describing the electrotonic properties of the real pyramidal cells. Each interneuron is modeled as a single compartment. Ion channels were described using the Hodgkin-Huxley formalism and channel densities were adjusted to achieve the firing patterns observed experimentally (*Vanier and Bower, 1999*). The inputs from the olfactory bulb were simulated by 960 spike generators parameterized on real mitral cells (*Bhalla and Bower, 1997*).

The connectivity of the network is based on the convergence and divergence patterns among the piriform cortex neurons. The pyramidal cells have excitatory recurrent connections with other pyramidal cells. Also, pyramidal cells have excitatory connections with feedback interneurons –FB cells-that send inhibitory projections to the pyramidal cells. Pyramidal cells receive inputs from mitral cells through the lateral olfactory tract (LOT). A sniffing cycle is simulated by theta bursts of weak synchronizing shocks in the LOT and odors are simulated by synchronizing transiently specific populations of firing mitral cells. The LOT conduction velocity of 7 mm/ms (*Haberly, 1973*; *Ketchum and Haberly, 1993*) was reduced by half and one third to simulate the effects of myelin disruption (*Gutiérrez et al., 1995*).

LFPs were calculated using the currents flowing through each pyramidal cell compartment and the resistivity of the piriform tissue (*Bower and Beeman, 2007*; *Wilson and Bower, 1992*). The power spectral frequency of the network oscillations was calculated by the fast Fourier transform (FFT), and the peaks of the FFT power were taken within the theta (6 to 14 Hz) and beta bandwidths (15 to 30 Hz).

## Tetrode construction

Four tetrodes consisting of four polyimide-coated nichrome wires (diameter 12.5 um; Sandvik, Palm Coast, FL) gold plated to an impedance of 0.2–0.4 Mohms were connected to an EIB-16 interface board (Neuralynx, Bozeman, MT).

## Surgery for implantation of tetrodes

Mice were briefly exposed to isoflurane (2.5%) prior to i.p. injection of ketamine-xylazine(100 ug/g and 20 ug/g, respectively). Anesthesia was verified by an absent toe pinch and tail pinch response. The fur on the head was cleaned using 100% ethanol and trimmed with scissors. The eyes were protected with ophthalmic ointment. The mouse was secured into a stereotaxic device with ear bars with the body aligned flat. The skin above the scalp was sliced down midline from the midpoints of the orbits to the midpoints of the ears to expose the skull. Two screw holes were drilled in the skull above the parietal cortex and one of the screws served as ground. Another hole was drilled above the frontal cortex within a diagonal band that overlies the piriform cortex: 1.6 mm anterior to bregma, 2.3 mm from the midline to 0.2 mm anterior to bregma, 3.4 mm from the midline. Neural activity was monitored visually as the tetrode was gradually lowered at depths ranging from 3.6 to 4.1 mm. The tetrode was secured to the bone using dental acrylic and allowed to dry.

## Recording and data analysis

The recording setup is as described in (*Li et al., 2015*). The output of the electrodes/tetrodes was directed to a TDT (Alachua, FL) 1X gain headstage that was connected to an A-M Systems 3600 amplifier. The signal was amplified 1000 times before digitizing with a Data Translation Inc. (Marlboro, MA) DT3010 A/D card in a PC. Data were acquired at 24 kHz and low pass filtered at 3 kHz. Data acquisition was controlled with custom software written in MATLAB (MathWorks, Inc., Natick,

MA, RRID:SCR_001622)(*Restrepo, 2018*). Time-frequency power decomposition of the LFP was obtained by means of MATLAB's *spectrogram.m* function with a 1 s window and 90% overlap. To compare LFP power between genotypes, we utilized Mann-Whitney U-test with false discovery rate correction for multiple comparisons (*Curran-Everett, 2000*).

## Acknowledgements

Funding was provided by a pilot grant from the National Multiple Sclerosis Society and NIH grants NS25304 (WBM), DC00566 and DC014253 (DR), AG053690 (DS), DC012280 and NS099042 (EG), NS048154, and IBM/FAPESP 2016/18825-4. We would like to thank Nicole Arevalo, Georgia Buscaglia, Katherine Given, Sean Hickey, Cayla Jewett for assistance.

## Additional information

### Funding

| Funder | Grant reference number | Author |
|---|---|---|
| National Institutes of Health | NS25304 | Wendy B Macklin |
| National Multiple Sclerosis Society | | Wendy B Macklin |
| National Institutes of Health | DC00566 | Diego Restrepo |
| National Institutes of Health | DC014253 | Diego Restrepo |
| National Institutes of Health | AG053690 | Douglas Shepherd |
| National Institutes of Health | DC012280 | Elizabeth A Gould |
| National Institutes of Health | NS099042 | Elizabeth A Gould |

The funders had no role in study design, data collection and interpretation, or the decision to submit the work for publication.

### Author contributions

Elizabeth A Gould, Conceptualization, Data curation, Software, Formal analysis, Investigation, Methodology, Writing—original draft, Writing—review and editing; Nicolas Busquet, Software, Formal analysis, Investigation, Methodology, Writing—original draft, Writing—review and editing; Douglas Shepherd, Data curation, Software, Formal analysis, Methodology, Writing—original draft, Writing—review and editing; Robert M Dietz, Investigation, Methodology, Writing—review and editing; Paco S Herson, Methodology, Writing—review and editing; Fabio M Simoes de Souza, Software, Formal analysis, Methodology, Writing—review and editing; Anan Li, Investigation, Methodology, Writing—original draft, Writing—review and editing; Nicholas M George, Investigation, Visualization, Methodology; Diego Restrepo, Conceptualization, Data curation, Software, Formal analysis, Supervision, Funding acquisition, Methodology, Writing—original draft, Project administration, Writing—review and editing; Wendy B Macklin, Conceptualization, Data curation, Formal analysis, Supervision, Funding acquisition, Validation, Visualization, Methodology, Writing—original draft, Project administration, Writing—review and editing

### Author ORCIDs

Diego Restrepo (iD) http://orcid.org/0000-0002-4972-446X
Wendy B Macklin (iD) http://orcid.org/0000-0002-1252-0607

### Ethics

Animal experimentation: All animals used in this study were treated in accordance with the University of Colorado Animal Care and Use Committee guidelines. The University of Colorado Animal Care and Use Committee approved this study under protocol numbers 00270 and 00134.

Decision letter and Author response
Decision letter https://doi.org/10.7554/eLife.34783.035
Author response https://doi.org/10.7554/eLife.34783.036

## Additional files

Supplementary files
• Transparent reporting form
DOI: https://doi.org/10.7554/eLife.34783.032

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
