## [Decision Letter]

[Editors’ note: a previous version of this study was rejected after peer review, but the authors submitted for reconsideration. The first decision letter after peer review is shown below.]

Thank you for submitting your work entitled "Mild Myelin Disruption Elicits Early Alteration in Behavior and Proliferation in the Subventricular Zone" for consideration by *eLife*. Your article has been reviewed by two peer reviewers, and the evaluation has been overseen by a Reviewing Editor and a Senior Editor. The reviewers have opted to remain anonymous. Unfortunately, we regret to inform you that your manuscript will not be considered further.

Specifically, while we agree that the model presents an interesting means to detect early deficits in myelination that may later lead to behavioral problems, the reviewers felt that more work would be needed to link the cellular changes and observations with the behavioral phenotypes. In addition, it was felt that more in depth analyses were needed to demonstrate that new oligodendrocytes are being generated. It is important to rule out cell death of precursors and provide more evidence that subventricular zone changes occur before axon damage or degeneration. We refer you to the full comments of the reviewers below for details.

It is *eLife*'s policy to only invite a manuscript for resubmission when the needed changes can be accomplished within two months time. Although that leaves us no choice but to reject your paper at this juncture, if you feel you can address the reviewers' comments with additional experiments and analyses, we would be pleased to see a new version of the manuscript and would make an effort to return the paper to the same reviewers.

*Reviewer #1:*

The study by Gould et al. investigates a very important problem, i.e. the cellular response in neurogenic and gliogenic regions of the brain to axonal injury. In particular, the authors investigate the *Plp*-null mouse, in which axon disruption occurs as early as 2 months in cortical projection neurons. High-volume cellular quantification showed region-specific increase in oligodendrocyte cell density in the olfactory bulb and rostral corpus callosum in adult age. Furthermore, analysis of the SVZ revealed a large increase in progenitor cells due to enhanced proliferation that preceded axonal injury. Finally, young adult *Plp*-null mice also displayed substantial behavioral alterations, which were associated with mild myelin disruption.

Overall, this is a very interesting report, which highlights the importance of mild demyelination in white matter function. Furthermore, since mutations in the *Plp* gene or gene duplications lead to several dysmyelinating diseases, this report also has translational implications. There are two major aspects of the paper that will require revisions: 1) the cellular analysis of the SVZ in the *Plp*-null mouse and the link between changes in the SVZ and enhanced oligodendrocyte production, and 2) the link between cellular changes – both in axons and in oligodendrocytes – and behavioral abnormalities.

• The authors claim that enhanced proliferation in the SVZ is generating new oligodendrocytes (OLs) in the OB and CC, but they haven't definitively proven this. They are relying on the specificity of the PLP-eGFP reporter for labeling only mature OLs. However, they previously published a paper showing expression of the PLP-eGFP reporter in ventricular-zone progenitors and migrating OPCs (Harlow et al., 2014).

To remedy this: the authors need to do the following:

1) Lineage tracing or viral tracing to label SVZ-derived progenitors and follow them to the OB and CC during these time windows. These SVZ-derived progenitors should be analyzed for expression of mature OLs (example: CC1).

2) Better confocal images showing lack of PLP-eGFP expression in SVZ progenitors (Figure 3—figure supplement 2). Need to indicate what region of the SVZ is being shown. The star annotation is doing nothing to help. Did they look at younger mice (P18) as well?

3) Co-stain for mature OL markers (CC1) in EdU labeling experiments. Are all the EdU+/PLP-eGFP+ cells in the CC and OB expressing CC1?

4) Co-stain the EdU+ cells in the SVZ for neuronal progenitor markers (i.e. Pax6) and glial makers (Olig2, NG2). Is there an increase in glial progenitors in the SVZ of PLP mutants? Or is there a non-specific increase in both neuronal and glial progenitors?

• The authors claim that SVZ activation occurs before noticeable axonal disruption, but only looked at a couple regions (CC and striatum). What about looking at the optic nerve at P18? If SVZ activation is occurring before axonal damage, what signals are promoting the SVZ response?

• The authors should discuss why there is a selective increase in OLs in the OB and the CC of the PLP-null mice. The authors also show enhanced reactive gliosis. This aspect and the potential role of reactive gliosis should also be discussed more extensively.

• Figure 4. This figure shows no significant changes in oligodendrocyte cell death in *Plp*-null mice CC and OB, as demonstrated by TUNEL assay. This figure is not described or discussed in the Results section. Furthermore, it would be important to perform some capase-3 immunostaining as a further validation to specifically show no changes in apoptosis.

• The cellular data should be better linked to the behavioral results. How do the increases in OLs in the OB and CC – and the axonal spheroids – potentially explain the behavioral phenotype?

• Throughout the paper, many of the experiments are done at different time points (P18, 2 months, 3 months, 4 months, 6 months). There is little rationale of why these ages were analyzed for different experiments. It would be important to have a summary figure or cartoon that indicates what cellular and behavioral changes were found in the *Plp*-null mice at each age. That way the readers can make sense of the progression of the phenotype.

*Reviewer #2:*

Dysregulation of myelin, the insulating sheath around axons critical for normal axon function, contributes to many neurological disorders. Even mild disruption can build to major cognitive and motor deficits later in life. In this study, Gould et al. demonstrate that in a genetic mouse model of mild myelin disruption, mice lacking proteolipid protein (PLP) exhibit a distinct proliferative response in progenitor cells specifically within the subventricular zone (SVZ), and increased oligodendrocyte density in corpus callosum and olfactory bulb. These abnormalities occur prior to axonal disruption (commonly seen in PLP models) and results in later cognitive and motor behavior deficits. PLP is a major structural component of myelin, and in this study Gould et al. use a *Plp*null mouse model to identify that well before late life myelin disruption, mice lacking PLP exhibit a significant increase in proliferation of precursor cells within the SVZ, and these cells preferentially become oligodendrocytes.

The authors demonstrated clearly the very exciting phenomenon of SVZ-specific proliferation of progenitor cells histologically. The authors use an interesting technique for high volume cell counting and support this with cell tracking data using timed EdU injections. However the effect this has on behavior is less clear. While there appears to be some effect on complex motor tasks, the anxiety data is contradictory.

1) The flow of text and figures are not clear and linear, particularly within the behavior section. Please organize the figures within the text to make the data clearer and easier to follow. For example, Figure 7 is essentially backwards to how the data is presented, and some behavioral tasks are described in the text after the data from them is presented.

2) In the puzzle box test, the claim is made that higher complexity tasks were impaired in both the 3M and 9M groups, however in the 3M group, though one sees deficits at the filled stage, there is no longer a deficit at the covered stage, the most complex portion of this task. Can you explain this difference? Why do the WT animals appear to have also struggled at this portion of the task specifically in this timepoint?

3) How were animals divided for this study? Were the same animals run on all tests, and tested at 3M and again at 9M? Or is this different sets of animals at each timepoint? What is the order in which the tests were run? It would be helpful to clarify the order of events for these experiments as multiple testing in behavior can change an animal's response, which could contribute to the confusing anxiety behavior data.

4) It would be interesting to track the animal's behavior from before the SVZ proliferation, or even immediately after, through the late time points. At the behavioral timepoints measured, these mice are already exhibiting massive axonal disruption. Is this behavior due specifically to axonal disruption and long term myelin deficits? Do these animals exhibit any of these motor deficits earlier due to developmental loss of PLP? Can behavior be run at an earlier timepoint (maybe at P18, to match with the cell counting data) to verify the specificity of behavior to later myelin disruption, rather than an effect of the genetic background?

5) Why was EM not run to verify myelin disruption at the timepoints measured?

6) In the Discussion, please expand upon the role of SVZ-derived progenitor cells in this model and its contribution in disease models. What do you hypothesize these cells are responding to? Further, since you see an increase of astrogliosis, do you suspect this is due more to inflammation?

---

## [Author Response]

[Editors’ note: the author responses to the first round of peer review follow.]

Reviewer #1:[…] Overall, this is a very interesting report, which highlights the importance of mild demyelination in white matter function. Furthermore, since mutations in the Plp gene or gene duplications lead to several dysmyelinating diseases, this report also has translational implications. There are two major aspects of the paper that will require revisions: 1) the cellular analysis of the SVZ in the Plp-null mouse and the link between changes in the SVZ and enhanced oligodendrocyte production.

We present two new experiments to address this question. In the new Figure 2—figure supplement 3 we show that the newly produced oligodendrocytes differentiated into Gst-pi+EDU+ mature oligodendrocytes in the OB and CC (Figure 2—figure supplement 3). In addition, in the new Figure 2—figure supplement 7 we show that Pax6+ cells expressing *Sox2*, a marker of progenitors was reduced, indicating a reduction in neural progenitors in the RMS, consistent with a complementary increase in oligodendrocyte progenitors. Finally, we revised the figure showing that the density of EDU+ neurons (NeuN+) and the percentage of NeuN+ cells that were EDU+ 3 weeks after EDU injection, were not altered in 4M *Plp*-null mice (Figure 2—figure supplement 6).

Thus, we have strengthened the characterization of the link from SVZ progenitors to mature oligodendrocytes. Future studies should use in utero electroporation or viral tracing to fully understand the changes in the SVZ lineage. However, in our opinion those experiments are beyond the scope of this manuscript.

2) The link between cellular changes – both in axons and in oligodendrocytes – and behavioral abnormalities.

We present new data linking cellular changes to behavioral abnormalities.

a) In Figure 4 we present measurements of the compound action potential (CAP) in CC showing a decrease in conduction velocity, and loss of the fast CAP peak at 2 and 6 months in the *Plp*-null.

b) In Figure 7—figure supplement 1 we present results of computational modeling of oscillations in piriform cortex that indicates that a decrease in conduction velocity in the lateral olfactory tract would lead to an increase in field potential oscillatory power in piriform cortex, as we had found in Figure 7.

• The authors claim that enhanced proliferation in the SVZ is generating new oligodendrocytes (OLs) in the OB and CC, but they haven't definitively proven this. They are relying on the specificity of the PLP-eGFP reporter for labeling only mature OLs. However, they previously published a paper showing expression of the PLP-eGFP reporter in ventricular-zone progenitors and migrating OPCs (Harlow et al., 2014).To remedy this: the authors need to do the following:1) Lineage tracing or viral tracing to label SVZ-derived progenitors and follow them to the OB and CC during these time windows. These SVZ-derived progenitors should be analyzed for expression of mature OLs (example: CC1).

We performed EDU labeling to quantify generation of newly generated mature oligodendrocytes identified with the mature marker Gst-pi+. We find that the newly produced oligodendrocytes differentiated into Gst-pi+ mature oligodendrocytes in the OB and CC (Figure 2—figure supplement 3).

In addition, we showed that Pax6+ cells expressing *Sox2*, a marker of progenitors was reduced, indicating a reduction in neural progenitors in the RMS (Figure 2—figure supplement 7), consistent with a complementary increase in oligodendrocyte progenitors.

2) Better confocal images showing lack of PLP-eGFP expression in SVZ progenitors (Figure 3—figure supplement 2). Need to indicate what region of the SVZ is being shown. The star annotation is doing nothing to help. Did they look at younger mice (P18) as well?

We revised the figure (now Figure 2—figure supplement 4) to make it clear that there is a lack of PLP-eGFP expression in SVZ progenitors. We did not look at P18 mice.

3) Co-stain for mature OL markers (CC1) in EdU labeling experiments. Are all the EdU+/PLP-eGFP+ cells in the CC and OB expressing CC1?

We co-stained EdU+ cells with the mature oligodendrocyte marker Gst-pi+. We find that the newly produced oligodendrocytes differentiated into Gst-pi+ mature oligodendrocytes in the OB and CC (Figure 2—figure supplement 3). In addition, we showed that Pax6+ cells expressing *Sox2*, a marker of progenitors was reduced, indicating a reduction in neural progenitors in the RMS (Figure 2—figure supplement 7).

4) Co-stain the EdU+ cells in the SVZ for neuronal progenitor markers (i.e. Pax6) and glial makers (Olig2, NG2). Is there an increase in glial progenitors in the SVZ of PLP mutants? Or is there a non-specific increase in both neuronal and glial progenitors?

We co-stained EdU+ cells with Pax6. Pax6+ cells were significantly reduced in the main and accessory olfactory bulb, indicating a specific loss of Pax6+ neurons (Figure 2—figure supplement 7). In addition, the Pax6+ cells expressing *Sox2*, a marker of progenitors was reduced, indicating a reduction in neural progenitors in the RMS (Figure 2—figure supplement 7). Therefore, the increase in oligodendrocytes occurs at the expense of the production of Pax6+ neurons. Please note that we also have data for co-staining of EdU+ and Olig2 (Figure 2).

• The authors claim that SVZ activation occurs before noticeable axonal disruption, but only looked at a couple regions (CC and striatum). What about looking at the optic nerve at P18?

We have included substantial new data in this revision of the manuscript. However, we had to limit the scope to a reasonable number of experiments, and we did not characterize axon disruption in the optic nerve.

If SVZ activation is occurring before axonal damage, what signals are promoting the SVZ response?

We do not know which signals are promoting the SVZ response. One of us (Macklin) was recently funded to study the involvement of mTOR signaling in oligodendrocyte differentiation. mTOR may be involved in the increase in Olig2 expression in SVZ.

• The authors should discuss why there is a selective increase in OLs in the OB and the CC of the PLP-null mice. The authors also show enhanced reactive gliosis. This aspect and the potential role of reactive gliosis should also be discussed more extensively.

We added a brief discussion of reactive gliosis: Finally, we observed region-specific reactive gliosis in the white matter of the 2M old *Plp*-null mouse (Figure 1—figure supplement 4). Whether the reactive gliosis is caused by *Plp* depletion, or is a reaction to axonal disruption is not known. Interestingly, reactive gliosis is found in the CC of animals treated with cuprizone where it is thought to be mediated by Toll-like receptors (Esser et al., 2017).

• Figure 4. This figure shows no significant changes in oligodendrocyte cell death in Plp-null mice CC and OB, as demonstrated by TUNEL assay. This figure is not described or discussed in the Results section. Furthermore, it would be important to perform some capase-3 immunostaining as a further validation to specifically show no changes in apoptosis.

Figure 4—figure supplement 2 is now presented in the Results. While further studies with caspase-3 immunostaining are interesting, our opinion is that this experiment is beyond what is already a comprehensive study of cellular and behavioral changes in *Plp*-null mice.

• The cellular data should be better linked to the behavioral results. How do the increases in OLs in the OB and CC – and the axonal spheroids – potentially explain the behavioral phenotype?

We present new data linking cellular to behavioral results:

a) New Figure 4 shows the results of measurement of conduction velocity in the corpus callosum for *Plp*-null and WT mice at three different ages: 18 days, 2M and 6M. We find that the fast N1 conduction velocity peak in the compound action potential (CAP) in the corpus callosum is present in 18 day old mice, but is absent in 2M and 6M *Plp*-null mice. In addition, we find that the 18 day old mice do not display gait anomalies, in contrast with the older mice that display overstepping of the hind leg (Results subsection “*Plp*-null mice display subtle motor deficits”). These data provide a functional link between the changes in oligodendrocyte myelination/axonal disruption to problems in motor coordination.

b) We modeled oscillatory activity in piriform cortex showing that a decrease in conduction velocity in the lateral olfactory tract is expected to cause an increase in oscillatory power in piriform cortex (new Figure 7—figure supplement 1). This observation provides a computational link between mild demyelination in the LOT and the increase in power we had observed in the previous version of the manuscript (Figure 7).

• Throughout the paper, many of the experiments are done at different time points (P18, 2 months, 3 months, 4 months, 6 months). There is little rationale of why these ages were analyzed for different experiments. It would be important to have a summary figure or cartoon that indicates what cellular and behavioral changes were found in the Plp-null mice at each age. That way the readers can make sense of the progression of the phenotype.

Thank you for the suggestion. We have generated a summary Figure 8 for the discussion that shows when the cellular, functional and behavioral changes take place.

Reviewer #2:Dysregulation of myelin, the insulating sheath around axons critical for normal axon function, contributes to many neurological disorders. Even mild disruption can build to major cognitive and motor deficits later in life. In this study, Gould et al. demonstrate that in a genetic mouse model of mild myelin disruption, mice lacking proteolipid protein (PLP) exhibit a distinct proliferative response in progenitor cells specifically within the subventricular zone (SVZ), and increased oligodendrocyte density in corpus callosum and olfactory bulb. These abnormalities occur prior to axonal disruption (commonly seen in PLP models) and results in later cognitive and motor behavior deficits. PLP is a major structural component of myelin, and in this study Gould et al. use a Plp-null mouse model to identify that well before late life myelin disruption, mice lacking PLP exhibit a significant increase in proliferation of precursor cells within the SVZ, and these cells preferentially become oligodendrocytes.The authors demonstrated clearly the very exciting phenomenon of SVZ-specific proliferation of progenitor cells histologically. The authors use an interesting technique for high volume cell counting and support this with cell tracking data using timed EdU injections.

We are pleased that the reviewer finds exciting that we demonstrated clearly the phenomenon of an increase in SVZ-specific proliferation in *Plp*-null mice. In addition, the reviewer finds interesting the use of the cleared tissue digital scanned light-sheet microscopy (C-DSLM) for high volume cell counting supported by complementary data obtained using timed EdU injections. A manuscript describing the development of the CDSLM, that automatically corrects for changes in refraction index in partially cleared tissue allowing faithful light sheet imaging of *Plp*-eGFP is described in a recently published manuscript that is now referenced in the Materials and methods (Ryan et al., 2017).

However the effect this has on behavior is less clear. While there appears to be some effect on complex motor tasks, the anxiety data is contradictory.

We have rearranged the presentation of the behavioral data and we present new data to make it clear for which subset of behaviors there are clear differences between WT and *Plp*-null. The select behavioral effects are clear and are backed up by solid data. We respond to each specific point below.

1) The flow of text and figures are not clear and linear, particularly within the behavior section. Please organize the figures within the text to make the data clearer and easier to follow. For example, Figure 7 is essentially backwards to how the data is presented, and some behavioral tasks are described in the text after the data from them is presented.

We agree. The flow of the text and figures was confusing, and this made it difficult to read the crucial section on behavior in the manuscript. We have thoroughly edited this section. The edits include a thematically-motivated split of Figure 7 into the new Figure 6 and Figure 6—figure supplements 1-5. We have also split the unwieldy table that was presented in Figure 2—figure supplement 6 into tables presented as supplements relevant to figures for each behavioral task. Furthermore, we present the behavioral analysis in a linear fashion starting with the motor task, that is closely related to changes in conduction velocity (CV) in the *Plp*-null and ending with olfactory function with the link to neural oscillations. The study of neural oscillations is now prefaced by a supplementary figure showing that circuit modeling of piriform cortex predicts an increase in oscillation power. Finally, part of the problem with linear cohesive presentation of the behavioral data was that we did not include a functional link from cellular changes to behavior. We now include a section on changes in axonal conduction velocity that provide a suitable bridge from cells to behavior (new Figure 4) followed by a preface to the behavioral characterization.

2) In the puzzle box test, the claim is made that higher complexity tasks were impaired in both the 3M and 9M groups, however in the 3M group, though one sees deficits at the filled stage, there is no longer a deficit at the covered stage, the most complex portion of this task. Can you explain this difference? Why do the WT animals appear to have also struggled at this portion of the task specifically in this timepoint?

The puzzle box test was designed with stages that involve on purpose different problem-solving tasks. Problem solving in each stage depends on learning at previous stages skills relevant to the present stage and problem solving and remembering previously learned skills in the present stage. The reviewer is correct that the 3M old group showed a deficit only in the filled stage. Why the 3M old did not have a subsequent deficit in the covered stage is not known, but it is likely due to skill learning in the previous stage. The wording has been modified to make it clear that the 3M old only had a deficit in the filled stage. Furthermore, it is not surprising that the WT mice struggle with transition to the last, most complex portion of the test.

3) How were animals divided for this study? Were the same animals run on all tests, and tested at 3M and again at 9M? Or is this different sets of animals at each timepoint? What is the order in which the tests were run? It would be helpful to clarify the order of events for these experiments as multiple testing in behavior can change an animal's response, which could contribute to the confusing anxiety behavior data.

We used the same cohort throughout when possible. However, in some cases we used different animals for different ages because of a constrained time frame (such as the period between receipt of the reviews and resubmission of the manuscript). Because of this we were not always able to perform tasks at different ages using the same mice. We have added this statement to the section on animal use: “Whenever possible, behavioral experiments at different ages were performed with the same cohort of animals.”

4) It would be interesting to track the animal's behavior from before the SVZ proliferation, or even immediately after, through the late time points. At the behavioral timepoints measured, these mice are already exhibiting massive axonal disruption. Is this behavior due specifically to axonal disruption and long term myelin deficits? Do these animals exhibit any of these motor deficits earlier due to developmental loss of PLP? Can behavior be run at an earlier timepoint (maybe at P18, to match with the cell counting data) to verify the specificity of behavior to later myelin disruption, rather than an effect of the genetic background?

This is an interesting comment. Please note that a substantial fraction of these behavioral tasks is not suitable for younger animals. To answer this question, we chose to perform gait analysis in 18-day-old mice to gain information on changes in motor coordination.

As stated in the Results (subsection “*Plp*-null mice display subtle motor deficits”), in contrast with the older mice, we do not find differences in gait between 18-day-old WT and *Plp*-null mice. In addition, we measured the conduction velocity of CC axons. We find that the fast conduction N1 peak in the compound action potential is lost in 2M and 6M *Plp*-null mice, but N1 is not absent in the 18 day old mice (New Figure 4). Thus, SVZ activation and an increase in the number of oligodendrocytes occurs before changes in axonal swelling and the motor coordination problem.

5) Why was EM not run to verify myelin disruption at the timepoints measured?

EM was not run because there are published EM studies showing myelin disruption in *Plp*-null mice (references: Rosenbluth, Nave, Mierzwa, and Schiff 2006, Rosenbluth et al., 1996). In Author response image 1 we show electron micrographs showing myelin disruption in 2mo old *Plp*-null mice. These images come from tissue that was first prepared for physiology, then post-fixed for EM. We are treating them as preliminary data because the animals did not undergo full EM fixation before imaging. Controls and proper EM fixations are necessary to fully replicate previous studies, though our preliminary images are similar to the published literature. Because there are published studies and a full EM replication would take a substantial amount of time, we would prefer not to show these preliminary EM micrographs.

Citations:

Rosenbluth, J., Nave, K. A., Mierzwa, A., and Schiff, R. (2006). Subtle myelin defects in PLP-null mice. Glia, *54*(3), 172-182.

Rosenbluth, J., Stoffel, W., and Schiff, R. (1996). Myelin structure in proteolipid protein (PLP)-null mouse spinal cord. Journal of Comparative Neurology, 371(2), 336-344.

6) In the Discussion, please expand upon the role of SVZ-derived progenitor cells in this model and its contribution in disease models. What do you hypothesize these cells are responding to? Further, since you see an increase of astrogliosis, do you suspect this is due more to inflammation?

We modified the Discussion and we added a summary figure (Figure 8). We added a brief discussion of reactive gliosis: “Finally, we observed region-specific reactive gliosis in the white matter of the 2M old *Plp*-null mouse (Figure 1—figure supplement 4). Whether the reactive gliosis is caused by *Plp* depletion, or is a reaction to axonal disruption is not known. Interestingly, reactive gliosis is found in the CC of animals treated with cuprizone where it is thought to be mediated by Toll-like receptors (Esser et al., 2017).” Finally, at the end of the Discussion we speculate on translational application of compensatory changes in circuit oscillations and SVZ and we cite the recent review of Alameida and Lyons relevant to our findings.